# LEVERAGING HIERARCHICAL FEATURE SHARING FOR EFFICIENT DATASET CONDENSATION

## ABSTRACT

Given a real-world dataset, data condensation (DC) aims to synthesize a significantly smaller dataset that captures the knowledge of this dataset for model training with high performance. Recent works propose to enhance DC with data parameterization, which condenses data into parameterized data containers rather than pixel space. The intuition behind data parameterization is to encode *shared features* of images to avoid additional storage costs. In this paper, we recognize that images share common features in a hierarchical way due to the inherent hierarchical structure of the classification system, which is overlooked by current data parameterization methods. To better align DC with this hierarchical nature and encourage more efficient information sharing inside data containers, we propose a novel data parameterization architecture, *Hierarchical Memory Network (HMN)*. HMN stores condensed data in a three-tier structure, representing the dataset-level, class-level, and instance-level features. Another helpful property of the hierarchical architecture is that HMN naturally ensures good independence among images despite achieving information sharing. This enables instance-level pruning for HMN to reduce redundant information, thereby further minimizing redundancy and enhancing performance. We evaluate HMN on five public datasets (SVHN, CIFAR10 / 100, Tiny-ImageNet, and ImageNet-10) and compare HMN with nine DC baselines. The evaluation shows that our proposed method outperforms all baselines, even when trained with a batch-based loss consuming less GPU memory.

## 1 INTRODUCTION

Data condensation (DC) Wang et al. (2018), also known as data distillation, has emerged as a valuable technique for compute-efficient deep learning Bartoldson et al. (2023); Zheng et al. (2020). It aims to synthesize a much smaller dataset while maintaining a comparable model performance to the case with full dataset training. Data condensation offers advantages in various applications, such as continual learning Rosasco et al. (2022); Sangermano et al. (2022), network architecture search Zhao & Bilen (2023b), and federated learning Song et al. (2022); Xiong et al. (2022). Because of the considerable practical utility of data condensation, substantial efforts Du et al. (2022); Nguyen et al. (2021; 2020); Shin et al. (2023); Cui et al. (2023) have been invested in improving its efficacy. Among them, *data parameterization* Liu et al. (2022); Kim et al. (2022) has been proposed, which condenses data into parameterized data containers instead of the pixel space. Those data parameterization methods usually generate more images given the same storage budget and improve data condensation performance. The intuition behind data parameterization methods is to encode *shared features* among images together to avoid additional storage costs.

Recognizing this shared feature insight, it's important to delve deeper into the inherent structure of these shared features in datasets. We notice that images share common features in a hierarchical way due to the inherent hierarchical structure of the classification system. Even if images differ in content, they can still share features at different hierarchical levels. For example, two images of cats can share common features specific to the cat class, but an image of a cat and another of a dog may still have shared features of the broader animal class. However, current data parameterization methods that adopt factorization to share features among images overlook this hierarchical nature of shared features in datasets. In this paper, to better align with this hierarchical nature and encourage more efficient information sharing inside data containers, we propose a novel data parameterization architecture, *Hierarchical Memory Network (HMN)*. As illustrated in Figure 1, an HMN comprises a

three-tier memory structure: *dataset-level memory*, *class-level memory*, and *instance-level memory*. Examples generated by HMNs share information via common dataset-level and class-level memories.

Another helpful property of the hierarchical architecture is that HMN naturally ensures good independence among images. We find that condensed datasets contain redundant data, indicating room for further improvement in data condensation by pruning redundant data. However, pruning redundant images for current data parameterization methods is challenging, since methods like HaBa Liu et al. (2022) and LinBa Deng & Russakovsky (2022) adopt factorization to achieve information sharing among images. Factorization leads to weights in data containers associated with multiple training images, which causes difficulty in pruning a specific image. Different from factorization-based methods, HMN naturally ensures better independence among images. Even though images generated by HMN share dataset-level and class-level memories, each generated image has its own instance-level memory. Thus, pruning redundant images to achieve better data efficiency can easily be done by pruning corresponding instance-level memories. We take advantage of this property of HMNs by first condensing a given dataset to a slightly over-budget HMN and then pruning the instance-level memories of redundant images to get back within allocated budgets.

We evaluate our proposed methods on four public datasets (SVHN, CIFAR10, CIFAR100, and Tiny-ImageNet) and compare HMN with the other eight baselines. The evaluation results show that, even when trained with a low GPU memory consumption batch-based loss, HMN still outperforms all baselines, including those using high GPU memory trajectory-based losses. For a fair comparison, we also compare HMN with other data parameterization baselines under the same loss. We find that HMN outperforms these baselines by a larger margin. For instance, HMN outperforms at least 3.7%/5.9%/2.4% than other data parameterization methods within 1/10/50 IPC (Image Per Class)[1] storage budgets when trained with the same loss on CIFAR10, respectively. Additionally, we also apply HMN to continual learning tasks. The evaluation results show that HMNs effectively improve the performance on continual learning.

To summarize, our contributions are as follows:

1. We propose a novel data parameterization method, Hierarchical Memory Network (HMN), comprising a three-tier memory structure: dataset-level, class-level, and instance-level.

2. We show that redundant data exist in condensed datasets. HMN inherently ensures good independence for generated images, facilitating the pruning of redundant images. We propose a pruning algorithm to reduce redundant information in HMNs.

3. We evaluate the performance of HMN on four public data and show that HMN outperforms eight SOTA baselines, even when we train HMNs with a batch-based loss consuming less GPU memory. We also compare HMN with other data parameterization baselines under the same loss. We find that HMN outperforms baselines by a larger margin. We thus believe that HMN provides a new baseline for exploring data condensation with limited GPU memory.

## 2 RELATED WORK

There are two main lines of approaches for improving data condensation: 1) designing better training losses and 2) increasing representation capability by data parameterization:

**Training losses for data condensation.** The underlying principle of data condensation is to optimize the synthetic dataset to exhibit a similar training behavior as the original dataset. There are two main types of training loss that are used to optimize synthetic datasets: 1) *trajectory-based loss* Wang et al. (2018); Cazenavette et al. (2022), and 2) *batch-based loss* Zhao & Bilen (2023b; 2021). Condensing using trajectory loss requires training the model on the synthetic dataset for multiple iterations while monitoring how the synthetic dataset updates the model parameters across iterations. For instance, MTT Cazenavette et al. (2022) employs the distance between model parameters of models trained on the synthetic dataset and those trained on the original dataset as the loss metric. In contrast, batch-based loss aims to minimize the difference between a batch of synthetic data and a batch of original data. Gradient matching Zhao et al. (2021); Lee et al. (2022); Jiang et al. (2022) calculates the distance between the gradients of a batch of condensed data and original data, while distribution

---

[1]IPC measures the equivalence of a tensor storage budget in terms of the number of images. For example, 1 IPC for CIFAR10 stands for: Pixels of an image * IPC * class = 3 * 32 * 32 * 1 * 10 = 30720 tensors. The same metric is also used in SOTA works Liu et al. (2022); Deng & Russakovsky (2022).

matching Zhao & Bilen (2023b) computes the distance between the embeddings of a batch of real data and original data. IDM Zhao et al. (2023) enhances distribution matching by improving the naive average embedding distribution matching. Since trajectory-based losses keep track of the long-term training behavior of synthetic datasets, trajectory-based losses generally show better empirical performance than batch-based losses. However, trajectory-based losses have considerably larger GPU memory consumption, potentially leading to scalability issues Cazenavette et al. (2022); Cui et al. (2022). *In this paper, we show that, equipped with HMN, a batch-based loss can also achieve comparable and even better performance than methods based on trajectory-based loss.*

**Data parameterization for data condensation.** Apart from training loss, *data parameterization* has been recently proposed as another approach to improve data condensation. Instead of utilizing independent images as data containers, recent works Deng & Russakovsky (2022); Liu et al. (2022); Kim et al. (2022) propose to use free parameters to store the condensed information. Those data parameterization methods usually generate more images given the same storage budget and improve data condensation performance by sharing information across different examples. HaBa Liu et al. (2022) and LinBa Deng & Russakovsky (2022) concurrently introduced factorization-based data containers to improve data condensation by sharing common information among images.

Some recent work Cazenavette et al. (2023); Zhao & Bilen (2022) explores generating condensed datasets with generative priors Brock et al. (2017); Chai et al. (2021a;b). For example, instead of synthesizing the condensed dataset from scratch, GLaD Cazenavette et al. (2023) assumes the existence of a well-trained generative model. We do not assume the availability of such a generative model thus this line of work is beyond the scope of this paper.

**Coreset Selection.** Coreset selection is another technique aimed at enhancing data efficiency Coleman et al. (2019); Xia et al. (2023); Li et al. (2023); Sener & Savarese (2017); Sorscher et al. (2022). Rather than generating a synthetic dataset, coreset selection identifies a representative subset from the original dataset. The majority of coreset selection methods select more important examples from datasets based on heuristic importance metrics. For instance, the area under the margin (AUM) (Pleiss et al., 2020) measures the data importance by accumulating output margin across training epochs. In the area of data condensation, coreset selection is used to select more representative data to initialize condensed data Cui et al. (2022); Liu et al. (2023).

## 3 METHODOLOGY

In this section, we present technical details on the proposed data condensation approach. In Section 3.1, we present the architecture design of our novel data container for condensation, Hierarchical Memory Network (HMN), to better align with the hierarchical nature of common feature sharing in datasets. In Section 3.2, we first study the data redundancy of datasets generated by data parameterization methods and then introduce our pruning algorithm on HMNs.

### 3.1 HIERARCHICAL MEMORY NETWORK (HMN)

Images naturally share features in a hierarchical way due to the inherent hierarchical structure of the classification system. For instance, two images of cats can share common features specific to the cat class, but an image of a cat and another of a dog may still have shared features of the broader animal class. To better align with the hierarchical nature of feature sharing in datasets, we propose a novel data parameterization approach, Hierarchical Memory Network (HMN). Our key insight for HMN is that images from the same class can share class-level common features, and images from different classes can share dataset-level common features. As shown in Figure 1, HMN is a three-tier hierarchical data container to store condensed information. Each tier comprises one or more memory tensors, and memory tensors are learnable parameters. The first tier is a dataset-level memory, $m^{(D)}$, which stores the dataset-level information shared among all images in the dataset. The second tier, the class-level memory, $m_c^{(C)}$, where $c$ is the class index. The class-level memories store class-level shared features. The number of class-level memories is equivalent to the number of classes in the dataset. The third tier stores the instance-level memory, $m_{c,i}^{(I)}$, where $c, i$ are the class index and instance index, respectively. The instance-level memories are designed to store unique information for each image. The number of instance-level memories determines the number of images the HMN generates for training. Besides the memory tensors, we also have feature extractors $f_i$ for each class and a uniform decoder $D$ to convert concatenated memory to images. Note that *both memory tensors and networks count for storage budget calculation.*

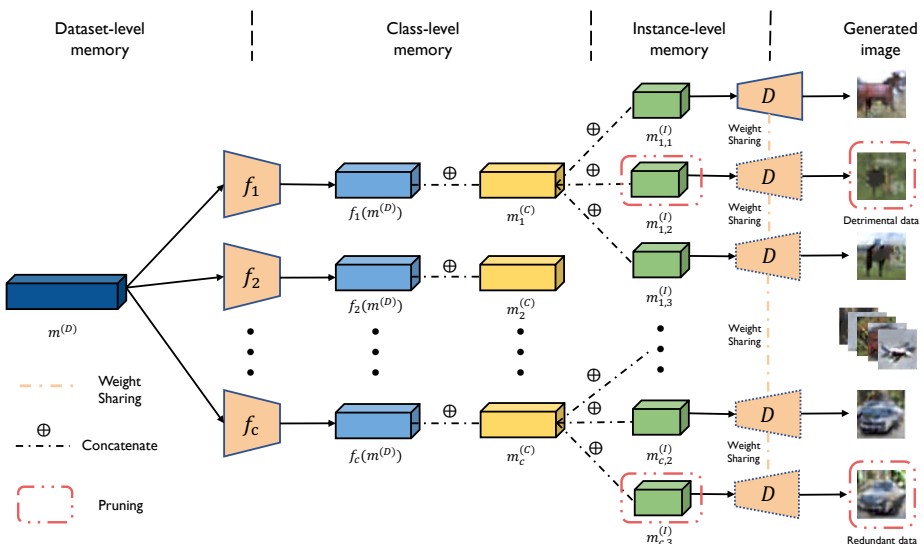

Figure 1: Illustration of Hierarchical Memory Network and pruning. HMN consists of three tiers of memories (which are learnable parameters). $f_i$ is the feature extractor for each class. $D$ is a uniform decoder to translate concatenated memories to images. When we identify redundant or detrimental images, the corresponding instance-level memories are pruned, as indicated by red boxes.

**Other design attempts.** In the preliminary stages of designing HMNs, we also considered applying feature extractors between $m_c^{(C)}$ and $m_{c,i}^{(I)}$, and attempted to use different decoders for each class to generate images. However, introducing such additional networks did not empirically improve performance. In some cases, it even causes performance drops. One explanation for these performance drops with an increased number of networks is overfitting: more parameters make a condensed dataset better fit the training data and specific model initialization but compromise the model's generalizability. Consequently, we decided to only apply feature extractors on the dataset-level memory and use a uniform decoder to generate images.

To generate an image for class $c$, we first adopt features extractor $f_c$ to extract features from the dataset-level memory [2]. This extraction is followed by a concatenation of these features with the class-level memory $m_c^{(C)}$ and instance-level memory $m_{c,i}^{(I)}$. The concatenated memory is then fed to a decoder $D$, which generates the image used for training. Formally, the $i$th generated image, $x_{c,i}$, in the class $c$ is generated by the following formula:

$$x_{c,i} = D([f_c(m^{(D)}); m_c^{(C)}; m_{c,i}^{(I)}]) \tag{1}$$

We treat the size of memories and the number of instance-level memories as hyperparameters for architecture design. We present design details in Appendix C, including the shape of memories, the number of generated images per class, architectures of feature extractors and decoder.

**Training loss.** HMN can be integrated with various training losses for data condensation. As discussed in Section 2, trajectory-based loss typically exhibits better empirical performance compared to batch-based loss, but it consumes more GPU memory, which may result in scalability issues. In this paper, to ensure better efficiency and scalability for data condensation, we employ gradient matching Kim et al. (2022), a batch-based loss, to condense information into HMNs. Given the original dataset $\mathcal{T}$, the initial model parameter distribution $P_{\theta_0}$, the distance function $D$, and loss function $\mathcal{L}$, gradient matching aims to synthesize a dataset $\mathcal{S}$ by solving the following optimization:

$$\min_{\mathcal{S}} \mathbf{E}_{\theta_0 \sim P_{\theta_0}} \left[ \sum_{t=0}^{T-1} D(\nabla_\theta \mathcal{L}(\theta_t, \mathcal{S}), \nabla_\theta \mathcal{L}(\theta_t, \mathcal{T})) \right], \tag{2}$$

where $\theta_t$ is learned from $\mathcal{T}$ based on $\theta_{t-1}$, and $t$ is the iteration number. In our scenario, the condensed dataset $\mathcal{S}$ is generated by an HMN denoted as $H$. In Section 4.2, our evaluation results show that our

---

[2]In some storage-limited settings, such as when storage budget is 1IPC, we utilize the identity function as $f_c$.

data condensation approach, even when employing a batch-based loss, achieves better performance than other DC baselines, including those that utilize high-memory trajectory-based losses.

## 3.2 DATA REDUNDANCY IN CONDENSED DATASETS AND POST-CONDENSATION PRUNING

In this part, we first show that data redundancy exists in condensed datasets in Section 3.2.1. Then, we propose a pruning algorithm on HMN to reduce such data redundancy in Section 3.2.2.

### 3.2.1 DATA REDUNDANCY IN CONDENSED DATASETS

Real-world datasets are shown to contain many redundant data Zheng et al. (2023); Pleiss et al. (2020); Toneva et al. (2018). Here, we show that such data redundancy also exists in condensed datasets. We use HaBa Liu et al. (2022) as an example. We first measure the difficulty of training images generated by HaBa with the area under the margin (AUM) Pleiss et al. (2020), a metric measuring data difficulty/importance. The margin for example $(\mathbf{x}, y)$ at training epoch $t$ is defined as:

$$M^{(t)}(\mathbf{x}, y) = z_y^{(t)}(\mathbf{x}) - \max_{i \neq y} z_i^{(t)}(\mathbf{x}), \tag{3}$$

where $z_i^{(t)}(\mathbf{x})$ is the prediction likelihood for class $i$ at training epoch $t$. AUM is the accumulated margin across all training epochs:

$$\mathbf{AUM}(\mathbf{x}, y) = \frac{1}{T} \sum_{t=1}^{T} M^{(t)}(\mathbf{x}, y). \tag{4}$$

A low AUM value indicates that examples are hard to learn. Those examples with lower AUM value are harder to learn, thus are thought to provide more information for training and are more importantToneva et al. (2018); Pleiss et al. (2020); Zheng et al. (2023). Then, as suggested in SOTA coreset selection work Toneva et al. (2018), we prune out the data with smaller importance (high AUM). The results of coreset selection on the dataset generated by HaBa for CIFAR10 10 IPC are presented in Table 1. We find that pruning up to 10% of the training examples does not hurt accuracy. This suggests that these 10% examples are redundant and can be pruned to save the storage budget.

Table 1: Coreset selection on the training dataset generated by HaBa on CIFAR10 10 IPC. The data with high AUM is pruned first.

| Pruning Rate | 0 | 10% | 20% | 30% | 40% |
|---|---|---|---|---|---|
| Accuracy (%) | 69.5 | 69.5 | 68.9 | 67.6 | 65.6 |

Pruning on generated datasets is straightforward, but pruning relevant weights in data containers can be challenging. SOTA well-performed data parameterization methods, like LinBa and HaBa, use factorization-based methods to generate images. Factorization-based methods use different combinations between basis vectors and decoders to share information, but this also creates interdependence among images, making prune specific images in data containers challenging.

A potential solution for pruning factorization-based data containers is to prune basis vectors in the data containers (each basis vector is used to generate multiple training images). However, we show that directly pruning these basis vectors can lead to removing important data. In Figure 2, we plot the importance rank distribution for training data generated by each basis vector. We observe that the difficulty/importance of images generated by the same basis vector can differ greatly. Thus, simply pruning a basis vector does not guarantee selective pruning of only the desired images.

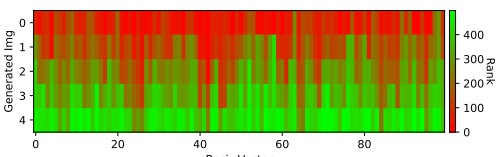

Figure 2: Rank distribution for different basis vectors in HaBa for CIFAR10 10 IPC. Each column in this figure represents the difficulty rank of images generated using the same basis vector. The color stands for the difficulty rank among all generated images. Green denotes easy-to-learn (less important) images, while red indicates hard-to-learn (more important) images.

Different from the factorization-based data condensation algorithm, HMN ensures good independence of each generated instance. As we can see in Figure 1, although generated images share information by using the same dataset-level and class-level memory, each generated image has its own instance-level memory, which allows us to prune redundant generated images by pruning corresponding instance-level memories (as illustrated by red dashed boxes in Figure 1).

### 3.2.2 OVER-BUDGET CONDENSATION AND POST-CONDENSATION PRUNING

To condense datasets with specific storage budgets and take advantage of the pruning property of HMN to further enhance data condensation, we propose to first condense data into over-budget HMNs, which exceed the storage budget by $p\%$ (which is a hyperparameter). Subsequently, we prune these HMNs to fit the allocated storage budget.

Inspired by recent coreset research Zheng et al. (2023) showing that pruning both easy and hard data leads to better coreset, we present a double-end pruning algorithm with an adaptive hard pruning rate to prune data adaptively for different storage budgets. As shown in Algorithm 1, given an over-budget HMN containing $k$ more generated images per class than allowed by the storage budget, we employ grid search to determine an appropriate hard pruning rate, denoted as $\beta$ (Line 4 to Line 12). We then prune $\lfloor \beta k \rfloor$ of the lowest AUM (hardest) examples and $k - \lfloor \beta k \rfloor$ of the highest AUM (easiest) examples by removing the corresponding instance-level memory for each class. The pruning is always class-balanced: the pruned HMNs generate the same number of examples for each class.

---

**Algorithm 1** Over-budget HMN Double-end Pruning

1: **Input:** Over-budget HMN: $H$; Over-budget images per class: $k$; $\beta$ search space $\mathcal{B}$.
2: Condensed dataset $\mathcal{S} \leftarrow H()$; $Acc_{best} = 0$; $\mathcal{S}_{best} = \emptyset$;
3: Calculate AUM for all examples in $\mathcal{S}$ based on Equation 4;
4: **for** $\beta$ **in** $\mathcal{B}$ **do**
5: $\quad \widetilde{\mathcal{S}} \leftarrow \mathcal{S}.clone()$;
6: $\quad$ Prune $\lfloor \beta k \rfloor$ of the lowest AUM examples for each class from $\widetilde{\mathcal{S}}$;
7: $\quad$ Prune $k - \lfloor \beta k \rfloor$ of the highest AUM examples for each class from $\widetilde{\mathcal{S}}$;
8: $\quad$ Retrain model $f$ on $\widetilde{\mathcal{S}}$;
9: $\quad Acc \leftarrow$ Test accuracy of the model $f$;
10: $\quad$ **if** $Acc > Acc_{best}$ **then**
11: $\quad\quad Acc_{best} = Acc$; $\widetilde{\mathcal{S}}_{best} = \widetilde{\mathcal{S}}$;
12: $\quad$ **end if**
13: **end for**
14: $\Delta\mathcal{S} = \mathcal{S} - \widetilde{\mathcal{S}}_{best}$;
15: $\widetilde{H} \leftarrow$ Prune corresponding instance-level memories in $H$ based on $\Delta\mathcal{S}$;
16: **Output:** Pruned in-budget network: $\widetilde{H}$.

---

Pruning in Algorithm 1 can introduce additional computational costs compared to the standard data condensation pipeline. However, we contend that, compared to the time required for data condensation, the pruning step requires a relatively small computation time. For example, while data condensation with HMNs for CIFAR10 1 IPC needs about 15 hours on a 2080TI GPU, the coreset selection on the condensed dataset only costs an additional 20 minutes.

## 4 EXPERIMENTS

In this section, we compare the performance of HMN to SOTA baselines in Section 4.2 and discuss the impacts of post-condensation pruning and HMN architecture design in Section 4.3. We also evaluate HMN on continual learning tasks in Section 4.4. Due to the page limitation, we include additional evaluation results in Appendix D: We compare the transferability of datasets generated by HMNs and other baselines in Appendix D.1. We then study the relationship between pruning rate and accuracy in Appendix D.3. Subsequently, we do data profiling and study the data redundancy on the condensed datasets synthesized by different DC methods in Appendix D.5. Lastly, we visualize the condensed training data generated by HMNs for different datasets in Appendix D.6.

### 4.1 EXPERIMENTAL SETTINGS

**Datasets and training settings** We evaluate our proposed method on four public datasets: CIFAR10, CIFAR100 (Krizhevsky et al., 2009), SVHN (Netzer et al., 2011), Tiny-ImageNet (Deng et al., 2009), and ImageNet-10 (Deng et al., 2009) under three different storage budgets: 1/10/50 IPC (For Tiny-ImageNet and ImageNet-10, due to the computation limitation, we conduct the evaluation on 1/10 IPC and 1 IPC, respectively). Following previous works Zhao & Bilen (2021); Liu et al. (2022); Deng & Russakovsky (2022), we select ConvNet, which contains three convolutional layers

Table 2: The performance (test accuracy %) comparison to state-of-the-art methods. The baseline method's accuracy is obtained from data presented in original papers or author-implemented repos. We label the methods using the trajectory-based training loss with a star (*). I-10 stands for ImageNet-10. We highlight the highest accuracy among all methods and methods with batch-based loss.

| Container | Dataset | CIFAR10 | | | CIFAR100 | | | SVHN | | | Tiny | | I-10 |
|---|---|---|---|---|---|---|---|---|---|---|---|---|---|
| | IPC | 1 | 10 | 50 | 1 | 10 | 50 | 1 | 10 | 50 | 1 | 10 | 1 |
| Image | DC | 28.3 ±0.5 | 44.9 ±0.5 | 53.9 ±0.5 | 12.8 ±0.3 | 25.2 ±0.3 | - | 31.2 ±1.4 | 76.1 ±0.6 | 82.3 ±0.3 | 4.6 ±0.6 | 11.2 ±1.6 | - |
| | DSA | 28.8 ±0.7 | 52.1 ±0.5 | 60.6 ±0.5 | 13.9 ±0.3 | 32.3 ±0.3 | 42.8 ±0.4 | 27.5 ±1.4 | 79.2 ±0.5 | 84.4 ±0.4 | 6.6 ±0.2 | 14.4 ±2.0 | - |
| | DM | 26.0 ±0.8 | 48.9 ±0.6 | 63.0 ±0.4 | 11.4 ±0.3 | 29.7 ±0.3 | 43.6 ±0.4 | - | - | - | 3.9 ±0.2 | 12.9 ±0.4 | - |
| | CAFE+DSA | 31.6 ±0.8 | 50.9 ±0.5 | 62.3 ±0.4 | 14.0 ±0.3 | 31.5 ±0.2 | 42.9 ±0.2 | 42.9 ±3.0 | 77.9 ±0.6 | 82.3 ±0.4 | - | - | - |
| | MTT* | 46.3 ±0.8 | 65.3 ±0.7 | 71.6 ±0.2 | 24.3 ±0.3 | 40.1 ±0.4 | 47.7 ±0.2 | 58.5 ±1.4 | 70.8 ±1.8 | 85.7 ±0.1 | 8.8 ±0.3 | 23.2 ±0.2 | - |
| | IDM | 45.6 ±0.7 | 58.6 ±0.1 | 67.5 ±0.1 | 20.1 ±0.3 | 45.1 ±0.1 | **50.0** ±0.2 | - | - | - | 10.1 ±0.2 | 21.9 ±0.2 | - |
| Data Parame- -terization | IDC | 50.0 ±0.4 | 67.5 ±0.5 | 74.5 ±0.2 | - | 45.1 | - | 68.5 | 87.5 | 90.1 | - | - | 60.4 |
| | HaBa* | 48.3 ±0.8 | 69.9 ±0.4 | 74.0 ±0.2 | 33.4 ±0.4 | 40.2 ±0.4 | 47.0 ±0.2 | 69.8 ±1.3 | 83.2 ±0.4 | 88.3 ±0.2 | - | - | - |
| | LinBa* | **66.4** ±0.4 | 71.2 ±0.4 | 73.6 ±0.5 | 34.0 ±0.4 | 42.9 ±0.7 | - | 87.3 ±0.1 | 89.1 ±0.2 | 89.5 ±0.2 | 16.0 ±0.7 | - | - |
| | HMN (Ours) | 65.7 ±0.3 | **73.7** ±0.2 | **76.9** ±0.2 | **36.3** ±0.2 | **45.4** ±0.2 | 48.5 ±0.2 | **87.4** ±0.2 | **90.0** ±0.1 | **91.2** ±0.1 | **19.4** ±0.1 | **24.4** ±0.1 | 64.6 |
| Entire Dataset | | 84.8 ±0.1 | | | 56.2 ±0.3 | | | 95.4 ±0.1 | | | 37.6 ±0.4 | | 90.8 |

followed by a pooling layer, as the network architecture for data condensation and classifier training. For the over-budget training and post-condensation, we first conduct a pruning study on HMNs in Section D.3, we observed that there is a pronounced decline in accuracy when the pruning rate exceeds 10%. Consequently, we select 10% as the over-budget rate for all settings. Nevertheless, we believe that this rate choice could be further explored, and other rate values could potentially further enhance the performance of HMNs. Due to space limits, we include more HMN architecture details, experimental settings, and additional implementation details in the supplementary material. All data condensation evaluation is repeated 3 times, and training on each HMN is repeated 10 times with different random seeds to calculate the mean with standard deviation.

**Baselines.** We compare our proposed method with eight baselines, which can be divided into two categories by data containers: **1) Image data container.** We use five recent works as the baseline: MTT Cazenavette et al. (2022) (as mentioned in Section 2). DC Zhao et al. (2021) and DSA Zhao & Bilen (2021) optimize condensed datasets by minimizing the distance between gradients calculated from a batch of condensed data and a batch of real data. DM Zhao & Bilen (2023b) aims to encourage condensed data to have a similar distribution to the original dataset in latent space. IDM Zhao et al. (2023) enhances distribution matching by improving the naive average embedding distribution matching. Finally, CAFE Wang et al. (2022) improves the distribution matching idea by layer-wise feature alignment. **2) Data parameterization.** We also compare our method with three SOTA data parameterization baselines. IDC Kim et al. (2022) enhances gradient matching loss calculation strategy and employs multi-formation functions to parameterize condensed data. HaBa Liu et al. (2022) and LinBa Deng & Russakovsky (2022) concurrently proposes factorization-based data parameterization to achieve information sharing among different generated images.

Besides grouping the methods by data containers, we also categorize those methods by the training losses used. As discussed in Section 2, there are two types of training loss: trajectory-based training loss and batch-based training loss. In Table 2, we highlight the methods using a trajectory-based loss with a star (*). In our HMN implementation, we condense our HMNs with gradient matching loss used in Kim et al. (2022), which is a low GPU memory consumption batch-based loss.

**The storage budget calculation.** As with other data parameterization techniques, our condensed data does not store images but rather model parameters. To ensure a fair comparison, we adopt the same setting of previous works Liu et al. (2022); Deng & Russakovsky (2022) and consider the total number of model parameters (*including both memory tensors and networks*) as the storage budget

(assuming that numbers are stored as floating-point values). For instance, the storage budget for CIFAR10 1 IPC is calculated as $32 * 32 * 3 * 1 * 10 = 30,720$. The HMN for CIFAR10 1 IPC always has an equal or lower number of parameters than this number.

## 4.2 Data Condensation Performance Comparison

We compare HMN with eight baselines on four different datasets (CIFA10, CIFAR100, SVHN, Tiny ImageNet, and ImageNet-10) in Table 2. We divide all methods into two categories by the type of data container formats: Image data container and data parameterization container. We also categorize all methods by the training loss. We use a star (*) to highlight the methods using a trajectory-based loss. The results presented in Table 2 show that HMN achieves comparable or better performance than all baselines. It is worth noting that HMN is trained with gradient matching, which is a low GPU memory loss. Two other well-performed data parameterization methods, HaBa and LinBa, are all trained with trajectory-based losses, consuming much larger GPU memory. These results show that batch-based loss can still achieve good performance with an effective data parameterization method and help address the memory issue of data condensation Cazenavette et al. (2022); Cui et al. (2022); Cazenavette et al. (2023). We believe that HMN provides a strong baseline for data condensation methods. We further study the memory consumed by different methods in Appendix D.2

**Data parameterization comparison with the same loss.** In addition to the end-to-end method comparison presented in Table 2, we also compare HMN with other data parameterization methods with the same training loss (gradient matching loss used by IDC) for a fairer comparison. The results are presented in Table 3. After replacing the trajectory-based loss used by HaBa and LinBa with a batch-based loss, there is a noticeable decline in accuracy (but HaBa and LinBa still outperform the image data container).[3] HMN outperforms other data parameterization by a larger margin when training with the same training loss, which indicates that HMN is a more effective data

Table 3: Accuracy (%) performance comparison to data containers with the same gradient matching training loss on CIFAR10. The evaluation results show that HMN outperforms all other data parameterization methods substantially.

| Data Container | 1 IPC | 10 IPC | 50 IPC |
|---|---|---|---|
| Image | 36.7 | 58.3 | 69.5 |
| IDC | 50.0 | 67.5 | 74.5 |
| HaBa | 48.5 | 61.8 | 72.4 |
| LinBa | 62.0 | 67.8 | 70.7 |
| HMN (Ours) | **65.7** | **73.7** | **76.9** |

parameterization method and can condense more information within the same storage budget. We also discussed the memory consumption of trajectory-loss on data parameterization in Appendix D.2.

## 4.3 Ablation Studies

**Ablation study on pruning.** In Table 4, we explore the performance of different pruning strategies applied to over-budget HMNs on the CIFAR10 dataset. The strategy termed "Prune easy" is widely employed in conventional coreset selection methods Coleman et al. (2019); Toneva et al. (2018); Paul et al. (2021); Xia et al. (2023), which typically prioritize pruning of easy examples containing more redundant information. "In-budget" refers to the process of directly condensing HMNs to fit the storage budgets, which does not need any further pruning. As shown in Table 4, our proposed pruning strategy (double-end) outperforms all other pruning strategies. We also observe that, as the storage budget increases, the accuracy improvement becomes larger compared to "in-budget" HMNs. We think this improvement

Table 4: Performance comparison on different pruning strategies on HMN. Double-end is the pruning strategy introduced in Algorithm 1. In-budget stands for HMNs are condensed within the allocated storage budget.

| IPC | 1 | 10 | 50 |
|---|---|---|---|
| Double-end | **65.7** | **73.7** | **76.9** |
| Prune easy | 65.3 | 73.1 | 76.6 |
| Random | 65.2 | 72.9 | 75.3 |
| In-budget | 65.1 | 73.2 | 75.4 |

is because a larger storage budget causes more redundancy in the condensed data Cui et al. (2022), which makes pruning reduce more redundancy in condensed datasets. Also, the performance gap between the "Prune easy" strategy and our pruning method is observed to narrow as the storage budget increases. This may be attributed to larger storage budgets for HMNs leading to more redundant easy examples. The "Prune easy" strategy can be a good alternative for pruning for large storage budgets.

---

[3]We do hyperparameter search for all data containers to choose the optimal setting.

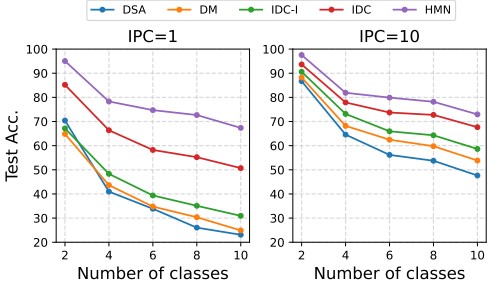 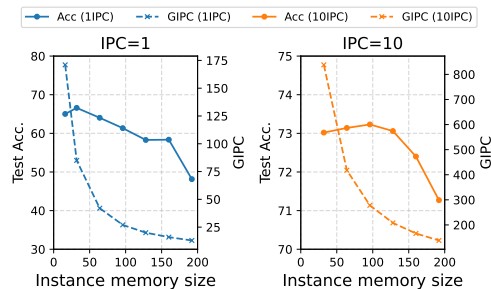

Figure 3: Continual learning evaluation on CI-FAR10. In the class incremental setting with 2 incoming classes per stage, HMN outperforms existing methods (including DSA, DM and IDC) under different storage budgets.

Figure 4: Instance-memory length vs. Accuracy for CIFAR10 HMNs with 1 IPC/10 IPC storage budgets. GIPC refers to the number of generated images per class. The solid and dashed curves represent the accuracy and GIPC, respectively.

**Instance-memory size v.s. Retrained model accuracy.** In an HMN, every generated image is associated with an independent instance-level memory, which constitutes the majority of the storage budget. Consequently, given a fixed storage budget, an increase in the instance-level memory results in a decrease in the number of generated images per class (GIPC). In Figure 4, we explore the interplay between the instance-memory size, the accuracy of the retrained model, and GIPC. Specifically, we modify the instance-level memory size of CIFAR10 HMNs for given storage budgets of 1 IPC and 10 IPC. (It should be noted that for this ablation study, we are condensing in-budget HMNs directly without employing any coreset selection on the condensed HMNs.)

From Figure 4, we observe that an increase in the instance-level memory size leads to a swift drop in GIPC, as each generated image consumes a larger portion of the storage budget. Moreover, we notice that both excessively small and large instance-level memory sizes negatively affect the accuracy of retrained models. Reduced instance-level memory size can result in each generated image encoding only a limited amount of information. This constraint can potentially deteriorate the quality of the generated images and negatively impact training performance. Conversely, while an enlarged instance-level memory size enhances the volume of information encoded in each image, it precipitously reduces GIPC. This reduction can compromise the diversity of generated images for training. For instance, with a 1IPC storage budget, an increase in the instance-level memory size, leading to a decrease in GIPC from 85 to 13, results in an accuracy drop from 65.1% to 48.2%.

### 4.4 CONTINUAL LEARNING PERFORMANCE COMPARISON

Following the same setting in DM Zhao & Bilen (2023a) and IDC Kim et al. (2022), we evaluate the effectiveness of HMN in an application scenario of continual learning Bang et al. (2021); Rebuffi et al. (2017b); Chaudhry et al. (2019). Specifically, we split the whole training phase into 5 stages, *i.e.* 2 classes per stage. At each stage, we condense the data currently available at this stage with ConvNet. As illustrated in Figure 3, evaluated on ConvNet models under the storage budget of both 1 IPC and 10 IPC, HMN obtains better performance compared with DSA Zhao & Bilen (2021), DM Zhao & Bilen (2023b), and IDC Kim et al. (2022). Particularly, in the low storage budget scenario, *i.e.* 1 IPC, the performance improvement brought by HMN is more significant, up to 16%. The results indicate that HMNs provide higher-quality condensed data and boost continual learning performance.

### 5 CONCLUSION

This paper introduces a novel data parameterization architecture, Hierarchical Memory Network (HMN), which is inspired by the hierarchical nature of common feature sharing in datasets. In contrast to previous data parameterization methods, HMN aligns more closely with this hierarchical nature of datasets. Additionally, we also show that redundant data exists in condensed datasets. Unlike previous data parameterization methods, although HMNs achieve information sharing among generated images, HMNs also naturally ensure good independence between generated images, which facilitates the pruning of data containers. The evaluation results on five public datasets show that HMN outperforms DC baselines, indicating that HMN is a more efficient architecture for DC.

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

## A   APPENDIX OVERVIEW

In this appendix, we provide more details on our experimental setting and additional evaluation results. In Section B, we discuss the computation cost caused by data parameterization. In Section C, we introduce the detailed setup of our experiment to improve the reproducibility of our paper. In Section D, we conduct additional studies on how the pruning rate influences the model performance. In addition, we visualize images generated by HMNs in Section D.6.

## B   DISCUSSION

While data parameterization methods demonstrate effective performance in data condensation, we show that generated images per class (GIPC) play an important role in data parameterization (as we discussed in Section 4.3). The payoff is that HMNs, along with other SOTA data parameterization methods Kim et al. (2022); Deng & Russakovsky (2022); Liu et al. (2022) invariably generate a higher quantity of images than those condensed and stored in pixel space with a specific storage budget, which may potentially escalate the cost of data condensation. A limitation of HMNs and other data parameterization methods is that determining the parameters of the data container to achieve high-quality data condensation can be computationally demanding. Besides, more generated images can lead to longer training time with condensed datasets. A key challenge and promising future direction is to explore approaches to reduce GIPC without affecting DC performance.

Another difference between data parameterization and conventional DC methods using images as data containers is that data parameterization methods need to generate images before training with condensed datasets. It is important to note that this additional step incurs only a minimal overhead, as it merely requires a single forward pass of HMNs. For example, on a 2080TI, the generation time for a 1 IPC, 10 IPC, and 50 IPC CIFAR10 HMN is 0.036s, 0.11s, and 0.52s, respectively (average number through 100 repeats).

## C   EXPERIMENT SETTING AND IMPLEMENTATION DETAILS

### C.1   HMN ARCHITECTURE DESIGN.

In this section, we introduce more details on the designs of the Hierarchical Memory Network (HMN) architecture, specifically tailored for various datasets and storage budgets. We first introduce the three-tier hierarchical memories incorporated within the network. Subsequently, we present the neural network designed to convert memory and decode memories into images utilized for training.

Table 5: The detailed three-tier memory settings. We use the same setting for CIFAR10 and SVHN. #Instance-level memory is the number of memory fitting the storage budget. #Instance-level memory (Over-budget) indicates the actual number of instance-level memory that we use for condensation, and we prune this number to #Instance-level memory after condensation. I-10 stands for ImageNet-10.

| Dataset | SVHN & CIFAR10 | | | CIFAR100 | | | Tiny | | I-10 |
|---|---|---|---|---|---|---|---|---|---|
| IPC | 1 | 10 | 50 | 1 | 10 | 50 | 1 | 10 | 1 |
| Dataset-level memory channels | 5 | 50 | 50 | 5 | 50 | 50 | 30 | 50 | 30 |
| Class-level memory channels | 3 | 30 | 30 | 3 | 30 | 30 | 20 | 30 | 25 |
| Instance-level memory channels | 2 | 6 | 8 | 2 | 8 | 14 | 4 | 10 | 8 |
| #Instance-level memory | 85 | 278 | 1168 | 93 | 219 | 673 | 42 | 185 | 125 |
| #Instance-level memory (Over-budget) | 93 | 306 | 1284 | 102 | 243 | 740 | 46 | 203 | 138 |

**Hierarchical memories.** HMNs consist of three-tier memories: dataset-level memory $m^{(D)}$, class-level memory $m_c^{(C)}$, and instance-level memory $m_{c,i}^{(I)}$, which are supposed to store different levels of features of datasets. Memories of HMNs for SVHN, CIFAR10, and CIFAR100 have a shape of (4, 4, Channels), and memories for Tiny ImageNet have a shape of (8, 8, Channels). Memories for ImageNet-10 have a shape of (12, 12, Channels). The number of channels is a hyper-parameter for different settings.

We present the detailed setting for the number of channels and the number of memories under different data condensation scenarios in Table 5. Besides the channels of memories, we also present the number of instance-level memories. Since each instance-level memory corresponds to a generated image, the number of instance-level of memories is the GPIC for an HMN. Every HMN has only one dataset-level memory, and the number of class-level memory is equal to the number of classes in the dataset. The number of instance-level memory for the over-budget class leads to an extra 10% storage budget cost, which will be pruned by post-condensation pruning.

**Decoders.** In addition to three-tier memories, each HMN has two types of networks: 1) A dataset-level memory feature extractor for each class; 2) A uniform decoder to convert memories to images for model training. *Dataset-level memory feature extractors* $f_c$ are used to extract features from the dataset-level memory for each class. For 1 IPC storage budget setting, we use the identity function as the feature extractor to save the storage budget. For 10 IPC and 50 IPC storage budget settings, the feature extractors consist of a single deconvolutional layer with the kernel with 1 kernel size and 40 output channels. *The uniform decoder $D$* is used to generate images for training. For ImageNet-10, the size of the generated image is (3, 96, 96). We use the bilinear interpolation to resize the generated images to (3, 224, 224). In this paper, we adopt a classic design of decoder for image generation, which consist of a series of deconvolutional layers and batch normalization layers: ConvTranspose(Channels of memory, 10, 4, 1, 2) $\rightarrow$ Batch Normalization $\rightarrow$ ConvTranspose(10, 6, 4, 1, 2) $\rightarrow$ Batch Normalization $\rightarrow$ ConvTranspose(6, 3, 4, 1, 2). The arguments for ConvTranspose is input-channels, output-channels, kernel size, padding, and stride, respectively. The "Channels of memory" is equal to the addition of the channels of the output of $f_c$, the class-level memory channels, and the instance-level memory channels. When we design the HMN architecture, we also tried the design with different decoders for different classes. However, we find that it experiences an overfitting issue and leads to worse empirical performance.

## C.2 TRAINING SETTINGS

**Baseline Settings** In this paper, we evaluate HMN on the same model and architecture and with the same IPC setting as the baselines for a fair comparison. For various baselines, we directly report the numbers represented in their papers. In general, as far as we can tell, the authors of various baselines chose reasonable hyperparameter settings, such as learning rate, learning rate schedule, batch size, etc. for their scheme. Sometimes the chosen settings differ. For instance, LinBa Deng & Russakovsky (2022) uses 0.1 as the learning rate, but HaBa Liu et al. (2022) uses 0.01 as the learning rate. In keeping with past work in this area, we accept such differences, since the goal of each scheme is to achieve the best accuracy for a given IPC setting. The settings that we found to be reasonable choices for HMN are described below. The metrics on which all schemes are being evaluated are the same: accuracy that the scheme is able to achieve for a given IPC setting.

**Data condensation.** We generally follow the guidance and settings from past work for the data condensation component of HMN. Following previous works Zhao & Bilen (2021); Liu et al. (2022); Deng & Russakovsky (2022), we select ConvNet, which contains three convolutional layers followed by a pooling layer, as the network architecture for data condensation and classifier training for all three datasets. For ImageNet-10, following previous work, we choose ResNet-AP (a four-layer ResNet) to condense HMNs. We employ gradient matching Kim et al. (2022); Liu et al. (2022), a batch-based loss with low GPU memory consumption, to condense information into HMNs. More specifically, our code is implemented based on IDC Liu et al. (2022). For all datasets, we set the number of inner iterations to 200 for gradient matching loss. The total number of training epochs for data condensation is 1000. We use the Adam optimizer ($\beta_1 = 0.9$ and $\beta_2 = 0.99$) with a 0.01 initial learning rate (0.02 initial learning rate for CIFAR100) for data condensation. The learning rate scheduler is the step learning rate scheduler, and the learning rate will time a factor of 0.1 at 600 and 800 epochs. We use the mean squared error loss for calculating the distance of gradients for CIFAR10 and SVHN, and use L1 loss for the CIFAR100 and Tiny ImageNet. To find the best hard pruning rate $\beta$ in Algorithm 1, we perform a grid search from 0 to 0.9 with a 0.1 step. All experiments are run on a combination of RTX2080TI, RTX3090, A40, and A100, depending on memory usage and availability.

**Model training with HMNs.** For CIFAR10, we train the model with datasets generated by HMNs for 2000, 2000, and 1000 epochs for 1 IPC, 10 IPC, and 50 IPC, respectively. We use the SGD optimizer (0.9 momentum and 0.0002 weight decay) with a 0.01 initial learning rate.

For CIFAR100, we train the model with datasets generated by HMNs for 500 epochs. We use the SGD optimizer (0.9 momentum and 0.0002 weight decay) with a 0.01 initial learning rate.

For SVHN, we train the model with datasets generated by HMNs for 1500, 1500, 700 epochs for 1 IPC, 10 IPC, and 50 IPC, respectively. We use the SGD optimizer (0.9 momentum and 0.0002 weight decay) with a 0.01 initial learning rate.

For both Tiny-ImageNet and ImageNet-10, we train the model with datasets generated by HMNs for 300 epochs for both 1 IPC and 10 IPC settings. We use the SGD optimizer (0.9 momentum and 0.0002 weight decay) with a 0.02 initial learning rate.

Similar to Liu et al. (2022), we use the DSA augmentation Zhao & Bilen (2021) and CutMix as data augmentation for data condensation and model training on HMNs. For HMN, for the learning rate scheduler, we use the cosine annealing learning rate scheduler Loshchilov & Hutter (2017) with a 0.0001 minimum learning rate. We preferred it over the multi-step learning rate scheduler primarily because the cosine annealing learning rate scheduler has fewer hyperparameters to choose. We also did an ablation study on the learning rate scheduler choice (see Appendix D.4) and did not find the choice of the learning rate scheduler to have a significant impact on the performance results.

**Continual learning.** Following the class incremental setting of Kim et al. (2022), we adopt distillation loss Li & Hoiem (2018) and train the model constantly by loading weights of the previous stage and expanding the output dimension of the last fully-connected layer Rebuffi et al. (2017a). Specifically, we use a ConvNet-3 model trained for 1000 epochs at each stage, using SGD with a momentum of 0.9 and a weight decay of $5e - 4$. The learning rate is set to 0.01, and decays at epoch 600 and 800, with a decaying factor of 0.2.

# D    ADDITIONAL EVALUATION RESULTS

In this section, we present additional evaluation results to further demonstrate the efficacy of HMNs. We compare the transferability of datasets generated by HMNs and other baselines in Section D.1. We then study the relationship between pruning rate and accuracy in Section D.3. Subsequently, we do data profiling and study the data redundancy on the condensed datasets synthesized by different DC methods in Section D.5. Lastly, we visualize the condensed training data generated by HMNs for different datasets in Section D.6.

## D.1    CROSS-ARCHITECTURE TRANSFERABILITY

Table 6: Transferability (accuracy %) comparison to different model architectures. Due to the extremely long training time, we cannot reproduce the results on LinBa. Compared with IDC and HaBa, we find that HMN achieves better performance for all model architectures.

| IPC | 1 | | | 10 | | | 50 | | |
|-----|-----|-----|------|-----|-----|------|-----|-----|------|
| Method | HMN | IDC | HaBa | HMN | IDC | HaBa | HMN | IDC | HaBa |
| ConvNet | **65.7** | 50.0 | 48.3 | **73.7** | 67.5 | 69.9 | **76.9** | 74.5 | 74 |
| VGG16 | **58.5** | 28.7 | 34.1 | **64.3** | 43.1 | 53.8 | **70.2** | 57.9 | 61.1 |
| ResNet18 | **56.8** | 32.3 | 36.0 | **62.9** | 45.1 | 49.0 | **69.1** | 58.4 | 60.4 |
| DenseNet121 | **50.7** | 24.3 | 34.6 | **56.9** | 38.5 | 49.3 | **65.1** | 50.5 | 57.8 |

To investigate the generalizability of HMNs across different architectures, we utilized condensed HMNs to train other network architectures. Specifically, we condense HMNs with ConvNet, and the condensed HMNs are tested on VGG16, ResNet18, and DenseNet121. We compare our methods with two other data parameterization methods: IDC and HaBa. (Due to the extremely long training time, we are unable to reproduce the results in LinBa). The evaluation results on CIFAR10 are presented in Table 6. We find that HMNs consistently outperform other baselines. Of particular interest, we observe that VGG16 has a better performance than ResNet18 and DenseNet121. A potential explanation may lie in the architectural similarities between ConvNet and VGG16. Both architectures are primarily comprised of convolutional layers and lack skip connections.

## D.2 GPU MEMORY COMPARISON

As discussed in Section 4.1, GPU memory consumption can be very different depending on the training losses used. We compare the GPU memory used by HMN with two other well-performed data parameterization methods, HaBa and LinBa. As depicted by Table 7, HMN achieves better or comparable performance compared to HaBa and LinBa with much less GPU memory consumption. Specifically, LinBa is trained with BPTT with a very long trajectory, which leads to extremely large GPU memory consumption. LinBa official implementation offloads the GPU memory to CPU memory to address this issue. However, the context switch in memory offloading causes the training time to be intolerable. For example, LinBa needs about 14 days to condense a CIFAR10 1IPC dataset with a 2080TI, but using HMN with gradient matching only needs 15 hours to complete training on a 2080TI GPU.

Although this memory saving does not come from the design of HMN, our paper shows that batch-based loss can still achieve very good performance with a proper data parameterization method, which helps address the memory issue of data condensation Cazenavette et al. (2022); Cui et al. (2022); Cazenavette et al. (2023).

Combining an HMN with the trajectory-based loss may further improve the performance of an HMN-based approach. We leave that investigation to future work since using trajectory-based loss in practice remains challenging from a scalability perspective due to high memory needs. Training with a batch-based loss allows us to evaluate our methods on more complex datasets with reasonable resources and in a reasonable time.

Table 7: Performance and memory comparison between LinBa, HaBa, and HMN trained with suggested loss in corresponding papers on CIFAR10.

| IPC | Method | Loss | Acc. | Memory |
|---|---|---|---|---|
| 1 | HaBa | MTT | 48.3 | 3368M |
| | LinBa | BPTT | **66.4** | OOM |
| | HMN (Ours) | GM-IDC | 65.7 | **2680M** |
| 10 | HaBa | MTT | 69.9 | 11148M |
| | LinBa | BPTT | 71.2 | OOM |
| | HMN (Ours) | GM-IDC | **73.7** | **4540M** |
| 50 | HaBa | MTT | 74 | 48276M |
| | LinBa | BPTT | 73.6 | OOM |
| | HMN (Ours) | GM-IDC | **76.9** | **10426M** |

## D.3 PRUNING RATE V.S. ACCURACY

In this section, we examine the correlation between accuracy and pruning rates on HMNs. The evaluation results are presented in Figure 5. We observe that the accuracy drops more as the pruning rates increase, and our double-end pruning algorithm consistently outperforms random pruning. Furthermore, we observe that an increasing pruning rate results in a greater reduction in accuracy for HMNs with smaller storage budgets. For instance, when the pruning rate increases from 0 to 30%, models trained on the 1 IPC HMN experience a significant drop in accuracy, plunging from 66.2% to 62.2%. Conversely, models trained on the 50 IPC HMN exhibit a mere marginal decrease in accuracy, descending from 76.7% to 76.5% with the same increase in pruning rate. This discrepancy may be attributed to the fact that HMNs with larger storage budgets generate considerably more redundant data. Consequently, pruning such data does not significantly impair the training performance.

## D.4 ABLATION STUDY ON LEARNING RATE SCHEDULER

We also train the model with a multi-step learning rate scheduler on CIFAR10 datasets generated by HMNs and found the following hyperparameter settings for a multi-step learning rate scheduler to work well: (a) an initial learning rate of 0.1; (b) The learning rate is multiplied with a 0.1 learning rate decay at 0.3 * total epochs / 0.6 * total epochs / 0.9 * total epochs. As shown in Table 8, we find the difference due to the LR scheduler choice to be overall marginal, and the results with the multistep LR scheduler do not change the findings of our evaluation. Our primary reason for choosing the cosine annealing LR scheduler in our evaluation is that it has fewer hyperparameters to choose from compared to the multistep LR scheduler. The cosine annealing

Table 8: Accuracy (%) performance comparison on different LR scheduler on CIFAR10. The evaluation results show that the difference due to the LR scheduler choice is overall marginal.

| Data Container | 1 IPC | 10 IPC | 50 IPC |
|---|---|---|---|
| Multi-step | 65.7 | 73.4 | 76.8 |
| Cosine Annealing | 65.7 | 73.7 | 76.9 |

LR scheduler only requires selection of an initial learning rate and a minimum learning rate. Those settings are described in Appendix C.2.

## D.5 DATA PROFILING ON SOTA METHODS

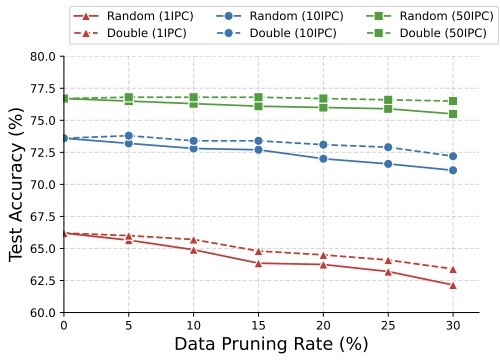

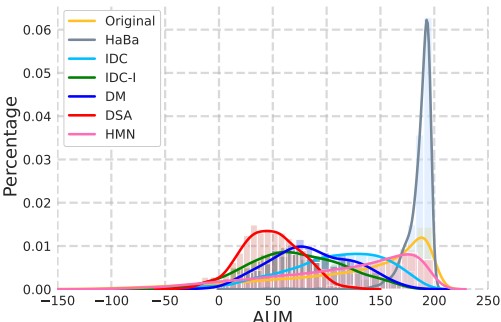

Figure 5: Relationship between pruning rates and accuracy on HMNs for CIFAR10. All HMNs are over-budget HMNs (10% extra). Different colors stand for different storage budgets. Solid lines stand for random pruning and dashed lines stand for double-end pruning.

Figure 6: The distribution of AUM of CIFAR10 training images synthesized by different approaches. Different colors denote different data condensation approaches. Data parameterization based methods have more redundant images.

Figure 6 illustrates the distribution of AUM of images synthesized by different data condensation approaches, as well as the original data, denoted as "Original". We calculate the AUM by training a ConvNet for 200 epochs. We observe that approaches (IDC-I Kim et al. (2022), DM Zhao & Bilen (2023a), and DSA Zhao & Bilen (2021)) that condense data into pixel space typically synthesize fewer images with a high AUM value. In contrast, methods that rely on data parameterization, such as HaBa Liu et al. (2022), IDC Kim et al. (2022), and HMN [4], tend to produce a higher number of high-aum images. Notably, a large portion of images generated by HaBa exhibit an AUM value approaching 200, indicating a significant amount of redundancy that could potentially be pruned for enhanced performance. However, due to its factorization-based design, HaBa precludes the pruning of individual images from its data containers, which limits the potential for efficiency improvements.

Moreover, we conduct a more detailed study on the images generated by HMNs. We calculate the AUM by training a ConvNet for 200 epochs. As shown in Figure 7, many examples possess negative AUM values, indicating that they are likely hard-to-learn, low-quality images that may negatively impact training. Moreover, a considerable number of examples demonstrate AUM values approximating 200, representing easy-to-learn examples that may contribute little to the training process. We also observe that an increased storage budget results in a higher proportion of easier examples. This could be a potential reason why data condensation performance degrades to random selection when the storage budget keeps increasing, which is observed in Cui et al. (2022): more storage budgets add more easy examples which only provide redundant information and do not contribute much to training. From Figure 7, we can derive two key insights: 1) condensed datasets contain easy examples (AUM close to 200) as well as hard examples (AUM with negative values), and 2) the proportion of easy examples varies depending on the storage budget.

Additionally, in Figure 8, we offer a visualization of images associated with the highest and lowest AUM values generated by an HMN. It is observable that images with low AUM values exhibit poor alignment with their corresponding labels, which may detrimentally impact the training process. Conversely, images corresponding to high AUM values depict a markedly improved alignment with their classes. However, these images may be overly similar, providing limited information to training.

---

[4]We did not evaluate LinBa due to its substantial time requirements.

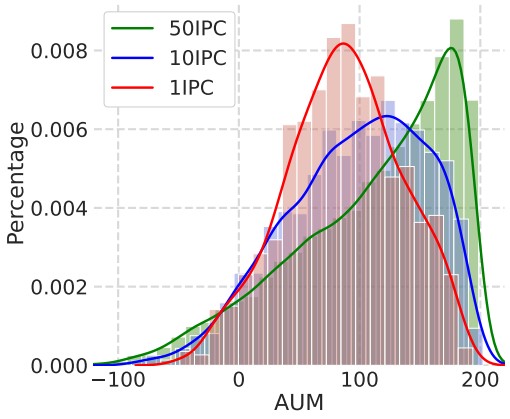 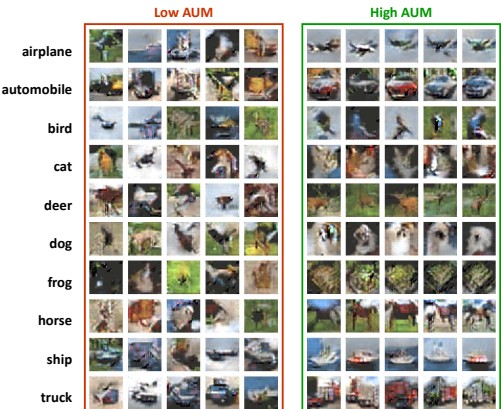

Figure 7: AUM distribution of images generated by HMNs for CIFAR10 with different storage budgets, denoted by different colors.

Figure 8: Visualization of the lowest and highest AUM examples generated by a 10 IPC HMN of CIFAR10. Each row represents a class.

### D.6 VISUALIZATION

To provide a better understanding of the images generated by HMNs, we visualize generated images with different AUM values on CIFAR10, CIFAR100, and SVHN with 1.1 IPC/11 IPC/55 IPC storage budgets in this section The visualization results are presented in the following images.

Similar to what we observe in Section 3.2 in the main paper, images with a high AUM value are better aligned with their respective labels. Conversely, images with a low AUM value typically exhibit low image quality or inconsistencies between their content and associated labels. For instance, in the visualizations of SVHNs (depicted in Figures 15 16 17), the numbers in the generated images with a high AUM value are readily identifiable, but content in the generated images with a low AUM value is hard to recognize. Those images are misaligned with their corresponding labels and can be detrimental to training. Pruning on those images can potentially improve training performance. Furthermore, we notice an enhancement in the quality of images generated by HMNs when more storage budgets are allocated. This improvement could be attributable to the fact that images generated by HMNs possess an enlarged instance-level memory, as indicated in Table 5. A larger instance-level memory stores additional information, thereby contributing to better image generation quality.

From the visualization, we also find that, unlike images generated by generative models, like GAN or diffusion models, images generated by HMNs do not exhibit comparably high quality. We would like to clarify that the goal of data condensation is not to generate high-quality images, but to generate images representing the training behavior of the original dataset. The training loss of data condensation can not guarantee the quality of the generated images.

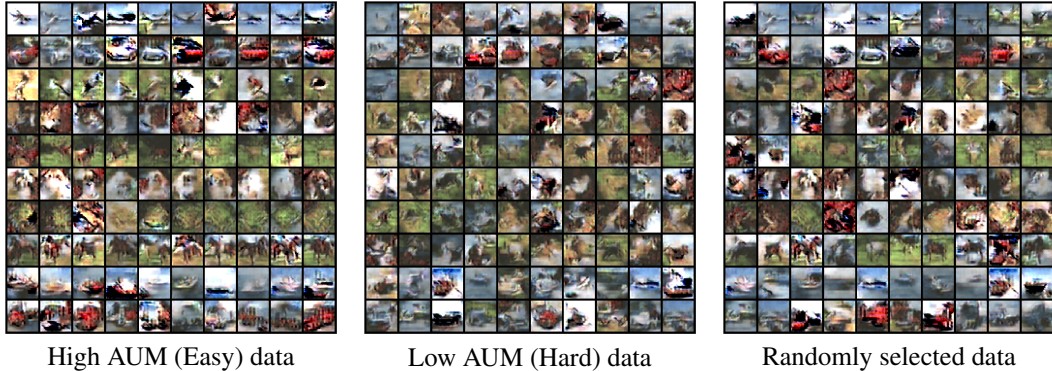

High AUM (Easy) data      Low AUM (Hard) data      Randomly selected data

Figure 9: Images generated by a CIFAR10 HMN with 1.1IPC storage budget. Images in each row are from the same class.

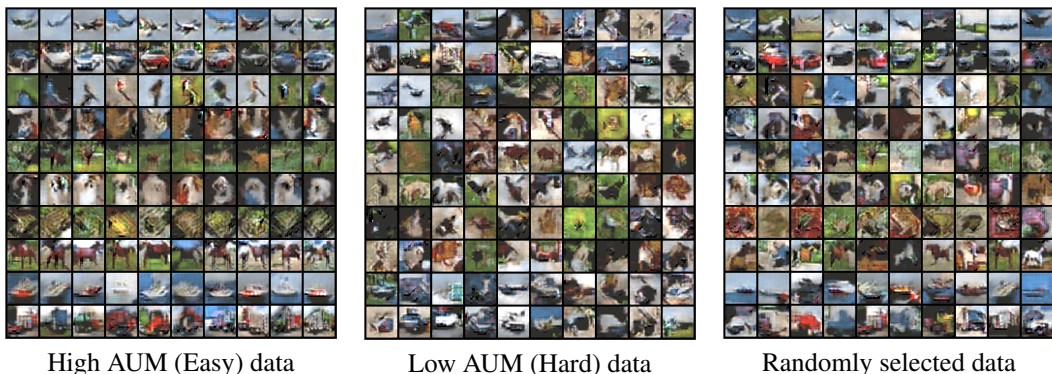

High AUM (Easy) data      Low AUM (Hard) data      Randomly selected data

Figure 10: Images generated by a CIFAR10 HMN with 11IPC storage budget. Images in each row are from the same class.

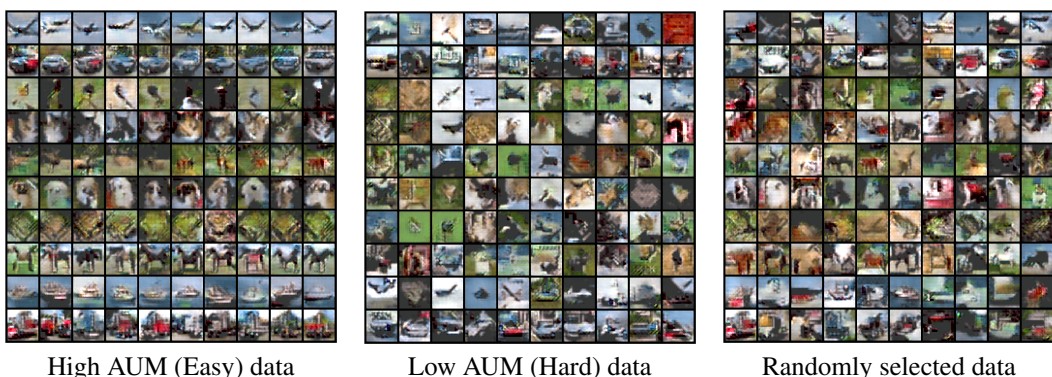

High AUM (Easy) data      Low AUM (Hard) data      Randomly selected data

Figure 11: Images generated by a CIFAR10 HMN with 55IPC storage budget. Images in each row are from the same class.

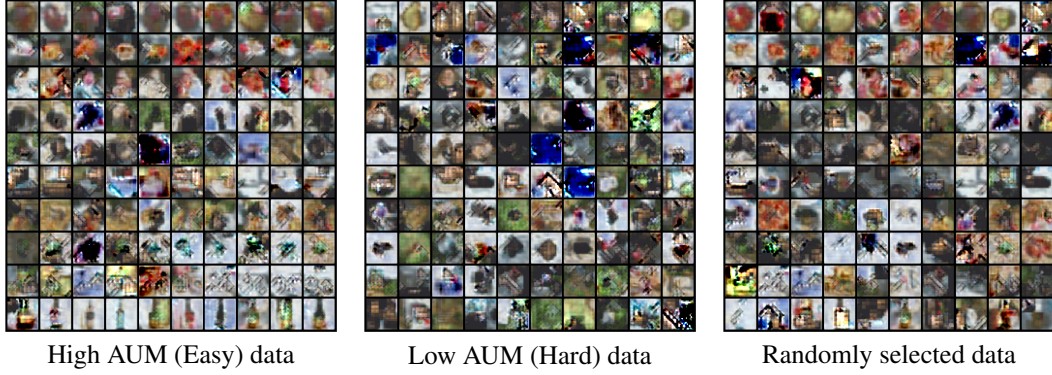

| High AUM (Easy) data | Low AUM (Hard) data | Randomly selected data |

Figure 12: Images generated by a CIFAR100 HMN with 1.1IPC storage budget. Images in each row are from the same class. We only visualize 10 classes with the smallest class number in the dataset.

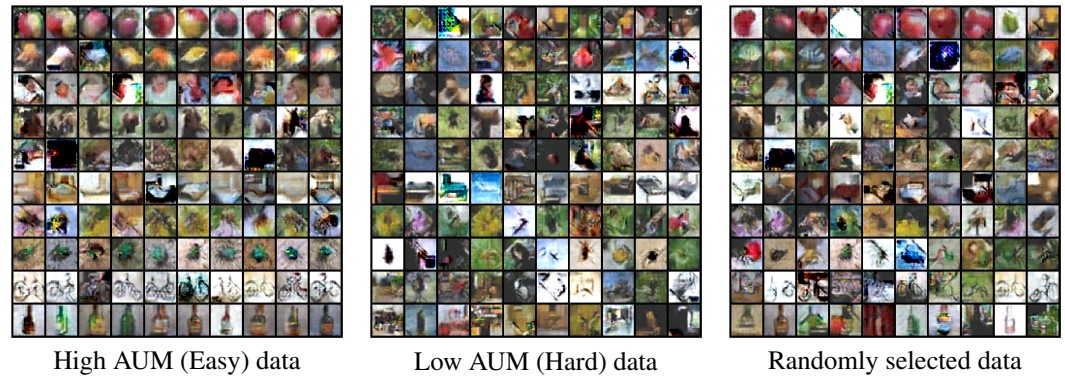

| High AUM (Easy) data | Low AUM (Hard) data | Randomly selected data |

Figure 13: Images generated by a CIFAR100 HMN with 11IPC storage budget. Images in each row are from the same class. We only visualize 10 classes with the smallest class number in the dataset.

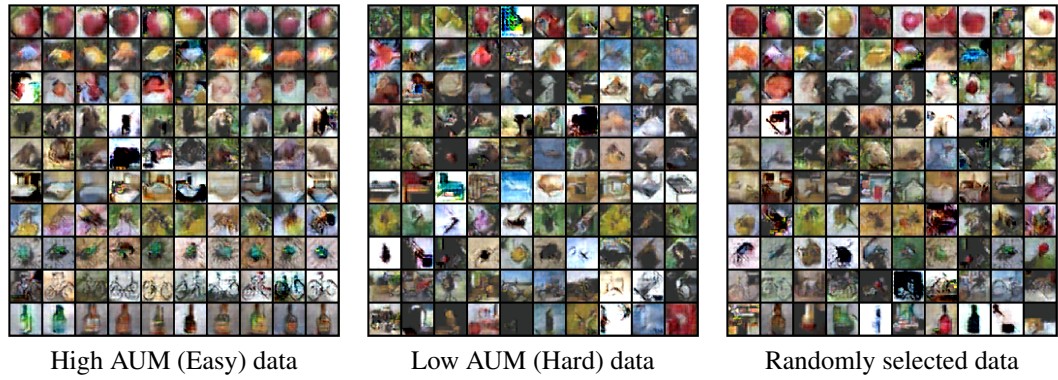

| High AUM (Easy) data | Low AUM (Hard) data | Randomly selected data |

Figure 14: Images generated by a CIFAR100 HMN with 55IPC storage budget. Images in each row are from the same class. We only visualize 10 classes with the smallest class number in the dataset.

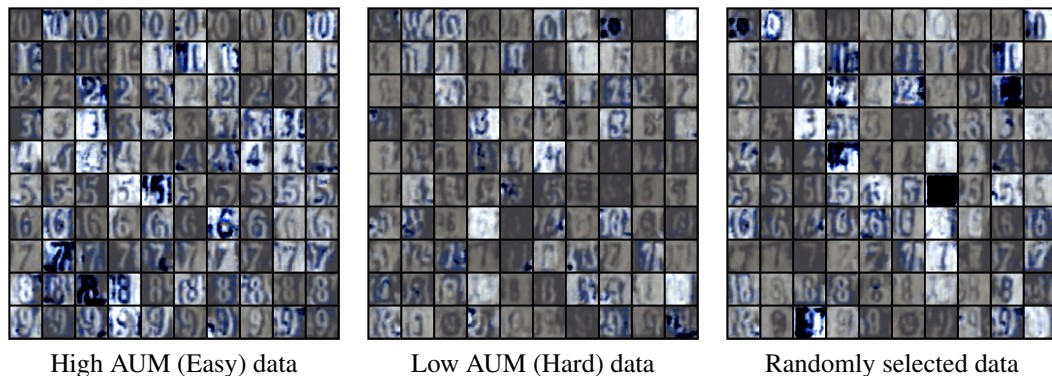

| High AUM (Easy) data | Low AUM (Hard) data | Randomly selected data |

Figure 15: Images generated by an SVHN HMN with 1.1IPC storage budget. Images in each row are from the same class. Images with a low aum value are not well-aligned with its label and can be harmful for the training.

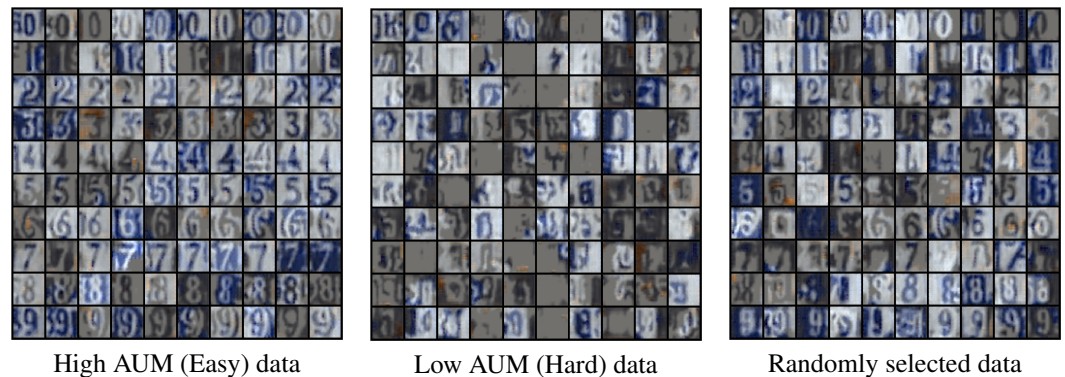

| High AUM (Easy) data | Low AUM (Hard) data | Randomly selected data |

Figure 16: Images generated by an SVHN HMN with 11IPC storage budget. Images in each row are from the same class.

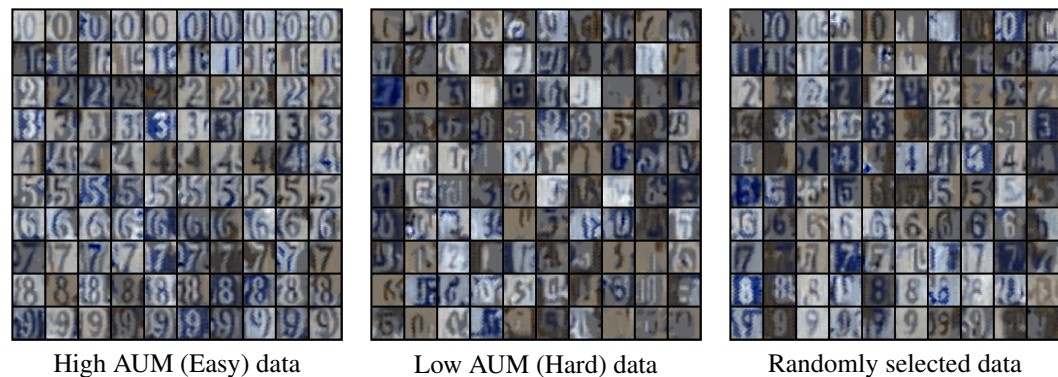

| High AUM (Easy) data | Low AUM (Hard) data | Randomly selected data |

Figure 17: Images generated by an SVHN HMN with 55IPC storage budget. Images in each row are from the same class.

