# OpenReview forum: "Leveraging Hierarchical Feature Sharing for Efficient Dataset Condensation"
_ICLR.cc/2024/Conference — Submitted to ICLR 2024_

### Official Review · Reviewer_UGPZ · 2023-10-28

**Soundness:** 3 good
**Presentation:** 3 good
**Contribution:** 3 good
**Rating:** 6
**Confidence:** 4

**Summary:**

This paper proposes a novel dataset condensation method following the data parameterization approach. The key idea is to use the newly designed Hierarchical Memory Network (HMN) to learn a three-tier representation of the condensed dataset. Experimental results demonstrate the effectiveness of the proposed method.

**Strengths:**

- The idea proposed is interesting and novel.
- The paper is well-written and easy to follow.

**Weaknesses:**

- It lacks a discussion and comparison to a very relevant SOTA method, i.e., IDM [1], which also addresses the scalability of data condensation and works in a low GPU memory cost scenario. Thus, it might be incorrect to argue that the proposed method is the "first method to achieve such good performance with a low GPU memory loss" and such statements should be revised.

[1] Zhao, G., Li, G., Qin, Y. and Yu, Y., 2023. Improved distribution matching for dataset condensation. In Proceedings of the IEEE/CVF Conference on Computer Vision and Pattern Recognition (pp. 7856-7865).

- The technical novelty is a bit thin. Specifically, the second contribution is relatively weak and more similar to a trick. It is interesting to know that over-budget generation and pruning may work, but this is more similar to remedying some of the shortcomings of the main idea rather than an independent contribution.

**Questions:**

Please see the weaknesses above.

---

> ### Author Response · Authors · 2023-11-17
> **Response to Reviewer #4 (UGPZ)**
>
> We thank Reviewer #4 (UGPZ) for the reviews and pointing out a missing related work. We address the raised questions and concerns below:
>
> **Q1:** It lacks a discussion and comparison to a very relevant SOTA method, i.e., IDM [1], which also addresses the scalability of data condensation and works in a low GPU memory cost scenario. Thus, it might be incorrect to argue that the proposed method is the "first method to achieve such good performance with a low GPU memory loss" and such statements should be revised.
>
> **A1:** We thank Reviewer #4 for pointing out a missing related work. As suggested by the reviewer, we have already added IDM to the Related Work section and added IDM as a new baseline in Table. 2 in the revised version. We also revised related statements in the paper. We would like to note that, *after adding IDM, our proposed method still achieves better or comparable performance than other baselines.*
>
> **Q2:** The technical novelty is a bit thin. Specifically, the second contribution is relatively weak and more similar to a trick. It is interesting to know that over-budget generation and pruning may work, but this is more similar to remedying some of the shortcomings of the main idea rather than an independent contribution.
>
> **A2:**  We would like to note that the major novelty in HMN’s design comes from its hierarchical architecture. HMN is the first data container that considers the hierarchical structure of the classification system, and HMN achieves a better empirical performance.  This makes us believe that *HMN provides certain novelty and contribution to the DC area.*
>
> We agree with the reviewer that the second contribution (HMN pruning part) is more like a favorable property of HMN. However, we note that not only HMN but also other data parameterization methods, like HaBa, also have data redundancy in the condensed data containers (as discussed in Section 3.2.1). Unlike the HMN approach, it is non-trivial to prune redundant generated examples in factorization-based data containers. HMN allows better independence between different generated images, which makes HMN a different data container that allows for further improving efficacy by pruning redundant information.
>
> -------------
>
> We thank Reviewer #4 again for pointing out an important missing related work and bringing the novelty of pruning for discussion. We hope our response addressed your concerns regarding the related work and novelty and hope you can kindly consider updating the rating once you feel satisfied with the changes. We are happy to have further discussion and any questions that might arise!

---

> ### Author Response · Authors · 2023-11-22
> **Thanks to Reviewer #4 (UGPZ) and the Response Summary on Major Concerns.**
>
> Dear Reviewer UGPZ,
>
> We greatly appreciate your time and efforts in reviewing our paper, and **we would like to briefly summarize the response to your major concerns for your convenience.** We also kindly refer you to our previous responses (both separate and general responses) for more detailed discussions.
>
> ---
>
> > **Q: Lack of discussion on a related data condensation paper, IDM.**
>
> **A:**  We have already added IDM to the Related Work section and added IDM as a new baseline in Table. 2 in the revised version. We would like to note that, after adding IDM, our proposed method still achieves better or comparable performance than SOTA methods.
>
> > **Q: The second contribution is relatively weak and more similar to a trick.**
>
> **A:** We would like to note that the major novelty in HMN’s design comes from its hierarchical architecture. HMN is the first data container that considers the hierarchical structure of the classification system. We agree with the reviewer that the second contribution (HMN pruning part) is more like a favorable property of HMN, but this property enables HMN to further improve efficacy by pruning redundant information, which is different from previous data parameterization methods.
>
> ----
>
> We have already revised the paper to include these points in the paper. **We also kindly refer you to our previous responses (both separate and general responses) for more detailed discussions.** We would like to check with you if our response and revised paper addressed your concerns and if we can provide more information to better address your concerns and clarify the claim in the paper!

---

### Official Review · Reviewer_RRfc · 2023-10-31

**Soundness:** 3 good
**Presentation:** 3 good
**Contribution:** 2 fair
**Rating:** 5
**Confidence:** 3

**Summary:**

The paper introduces a novel data parameterization architecture named Hierarchical Memory Network (HMN), which consists of dataset-level, class-level, and instance-level memory and a feature extractor and a decoder to better leverage the hierarchical nature of the image. Furthermore, the paper proposes instance-level pruning to remove redundant images to improve performance.

**Strengths:**

1. The paper proposes a novel framework for DC with data parameterization, instead of updating input images, they update the memory (i.e, dataset, class, and instance) and decoder.
2. Identify the redundancy in condensed images using AUM and suggest to first condense images with over-budget (p%) and then perform post-condensation pruning to remove redundant data.

**Weaknesses:**

Although the proposed method (including HMN and condensed over-budged & pruning) is novel, there are several weaknesses:

1. Lack of experiments on higher resolution such as 128x128 (Image Woof/Nette) or 224x224 (ImageNet-10/100, a subset from ImageNet, as done in IDC)
2. Lack of describing the memory in detail. For example, how does the memory look like?
3. The evaluation seems to be different from previous works. For example, HMN evaluates condensed images with a cosine annealing learning rate (LR) scheduler, while DSA or IDC uses a multistep LR scheduler. Thus, the comparison may not be fair.

**Questions:**

I have several questions:
1. What does the memory look like? How to initialize these memories, feature extractor, and decoder?
2. HMN stores parameters instead of images, this will incur extra resources for evaluation (to generate images). Is it correct?
3. How many generated images per class  (GIPC) used in Table 2?
4. In Table 2, the result of IDC on CIFAR-10 with 50 IPC is lower than in the paper (74.5 (IDC) vs 71.6 (this paper)). Is it a mistake?  On CIFAR-100, this paper reports the accuracy of IDC with 10 IPC on CIFAR-100 is 44.8 while on other papers such as DREAM, the accuracy is 45.1. Can the author double-check? The results of IDC (at 10 and 50 IPC) in Table 2, Table 3, and Table 6 are different. Can the author explain?
5. As shown in Figure 4 (right),  with an instance memory size near 100, around 600 images per class are generated. How many images per class (per epoch) are used for training in the evaluation phase? 600? If this is correct, the training time with condensed images is much longer than the conventional method where the total images per lass are set as the IPC. Can the author clarify this?

Although the proposed method is novel and seems promising, there are unclear parts regarding the GIPC, the memory, and fairness in evaluation comparison. Additionally, some reported results of IDC of this paper are different from those in the original paper without explanation. Thus, I give a borderline reject.

---

> ### Author Response · Authors · 2023-11-17
> **Response to Reviewer #3 (RRfc) (Part 1)**
>
> We thank Reviewer #3 (RRfc) for the reviews and suggestions to improve the paper. We address the raised questions and concerns below:
>
> **Q1:** Lack of experiments on higher resolution such as 128x128 (Image Woof/Nette) or 224x224 (ImageNet-10/100, a subset from ImageNet, as done in IDC)
>
> **A1:** We thank the reviewers for suggesting additional evaluation on a high-resolution dataset like ImageNet to better demonstrate the effectiveness of HMN. We now include the ImageNet-10 evaluation result in Table. 2 (Section 4.1) in the revised version for a more extensive comparison. We evaluated HMN on ImageNet-10 1 IPC setting and compared it with IDC (other baselines are too resource-intensive to run).  The evaluation results are summarized below:
>
> |      |  IDC |    HMN   |
> |:----:|:----:|:--------:|
> | 1 IPC | 60.4 | **64.6** |
>
> From the table, we find that HMN still achieves strong performance on ImageNet-10, outperforming IDC.
>
> **Q2:** Lack of describing the memory in detail. For example, how does the memory look like?
> What does the memory look like? How to initialize these memories, feature extractor, and decoder? (Q2)
>
> **A2:** All memories are 3-D tensors. For CIFAR10, CIFAR100, and SVHN, memories have a shape (4, 4, Channels). For Tiny-imageNet, memories have a shape (8, 8, Channels). All values in the tensor are initialized with standard normal distribution. We included more details about the memory design of HMN in Section 3.1 and Appendix C.1 in the revised version.
>
> **Q3:** The evaluation seems to be different from previous works. For example, HMN evaluates condensed images with a cosine annealing learning rate (LR) scheduler, while DSA or IDC uses a multistep LR scheduler. Thus, the comparison may not be fair.
>
> **A3:** We thank the reviewer for pointing this out and agree with the reviewer that the evaluation with a multistep LR scheduler can still be useful. We report below the performance of HMN with a multistep LR scheduler on the CIFAR10 dataset as follows.
>
> |      IPC       | 1  | 10   | 50  |
> |:----------------:|:----:|:-----:|-------|
> |    Multi-step    | 65.7 | 73.5  | 76.8  |
> | Cosine Annealing | 65.7 | 73.7  | 76.9  |
>
> We find the difference due to the LR scheduler choice to be marginal, and the results with the multistep LR scheduler do not change the findings and conclusions of our evaluation. Our primary reason for choosing the cosine annealing LR scheduler in our evaluation is that it has fewer hyperparameters compared to the multistep LR scheduler (like milestones to reduce learning rate, and learning rate decay factor). We added more details to Appendix C.2 and D.4 regarding our hyperparameter settings and the rationale for any differences.
>
> **Q4:** HMN stores parameters instead of images, this will incur extra resources for evaluation (to generate images). Is it correct?
>
> **A4:** The reviewer is right that all data parameterization methods need to generate training data before model training. We would like to clarify that this generation overhead is relatively small, because it just needs one forward pass of HMNs. For example, on a 2080TI, the generation time for a 1 IPC, 10 IPC, and 50 IPC CIFAR10 HMN is **0.036s, 0.11s, and 0.52s,** respectively (average number through 100 repeats).
>
> We thank the reviewer for pointing this out for discussion, and we have already added this to the discussion in Appendix B in the revised version.
>
> **Q5:** How many generated images per class (GIPC) used in Table 2?
>
> **A5:** We listed GIPC for each setting in Table 5 (Appendix C). Since each instance-level memory corresponds to a generated image, the row “#Instance-level memory” indicates the GIPC for each setting. We have also further revised the writing in Section 3.1 and Appendix C.1 to reduce unnecessary confusion about GIPC of HMNs.

---

> > ### Author Response · Authors · 2023-11-17
> > **Response to Reviewer #3 (RRfc) (Part 2)**
> >
> > **Q6:** As shown in Figure 4 (right), with an instance memory size near 100, around 600 images per class are generated. How many images per class (per epoch) are used for training in the evaluation phase? 600? If this is correct, the training time with condensed images is much longer than the conventional method where the total images per class are set as the IPC. Can the author clarify this?
> >
> > **A6:** We thank the reviewer for bringing the training time for discussion. We would like first to clarify that the dashed curve in Figure 4 is the GIPC curve, not the solid line. Thus, around 250 images per class generated (dashed line) at instance memory size near 100.
> >
> > The reviewer's larger point is valid in that a larger GIPC size can impact training time. We would like to point out that this is a common feature of data parameterization methods, including IDC, HaBa, LinBa, and HMN. They will all generate more images than conventional DC methods using images as data containers. For example, LinBa generates 115 images per class with the 1 IPC setting for CIFAR10. Although data parameterization methods generate more training images, data parameterization methods also achieve better accuracy. More GIPC can be treated as a tradeoff for better accuracy of data parameterization methods.  We note that this is a feature of all high-performing data parameterization methods, not only of HMNs. We leave the challenge of reducing GPIC for data parameterization methods, while retaining accuracy, for future exploration.
> >
> > As suggested by the reviewer, we have included those clarifications and discussions in Appendix B in the revised version.
> >
> > **Q7:** In Table 2, the result of IDC on CIFAR-10 with 50 IPC is lower than in the paper (74.5 (IDC) vs 71.6 (this paper)). Is it a mistake? On CIFAR-100, this paper reports the accuracy of IDC with 10 IPC on CIFAR-100 is 44.8 while on other papers such as DREAM, the accuracy is 45.1. Can the author double-check? The results of IDC (at 10 and 50 IPC) in Table 2, Table 3, and Table 6 are different. Can the author explain?
> >
> > **A7:** We appreciate Reviewer #3 for pointing this out. There is a typo on the evaluation result for the IDC CIFAR10 50 IPC setting. We have corrected the number (The original number is the number of MTT.). For the IDC CIFAR100 10 IPC setting, we use the data (44.8) from a data distillation survey paper [1]. However, we noticed that this number is different from the number reported in the IDC official repo [2]. We have updated this number to 45.1 in the revised version. After those modifications, the numbers in Tables 2 and 3 are corrected and the same as the author-reported numbers.
> >
> > The numbers in Table 6 were the results that we got by rerunning the IDC evaluation.  Some differences can be caused by randomness in training. We now also update the numbers in Table 6 to author-reported numbers to guarantee consistency in the paper. We thank the reviewer for helping us correct this.
> >
> > We would like to note that **the correction of those typos does not change the conclusion of our evaluation**. HMN still achieves better results.
> >
> > -------------
> >
> > We thank Reviewer #3 again for the insightful reviews. We appreciate the help in pointing out unclear points and typos of evaluation data in the paper, which indeed improves the quality of our paper. We hope that our response and revisions have successfully addressed your concerns and questions. We kindly hope that you can reconsider the rating in light of these updates. We are happy to have any further discussions or answer other questions that may arise, as we greatly value your feedback in this process.
> >
> >
> > [1] Sachdeva, Noveen, and Julian McAuley. "Data distillation: A survey." arXiv preprint arXiv:2301.04272 (2023).
> >
> > [2] https://github.com/snu-mllab/Efficient-Dataset-Condensation/

---

> ### Author Response · Authors · 2023-11-22
> **Thanks to Reviewer #3 (RRfc) and the Response Summary on Major Concerns.**
>
> Dear Reviewer RRfc,
>
> We greatly appreciate your time and efforts in reviewing our paper, and **we would like to briefly summarize the response to your major concerns for your convenience.** We also kindly refer you to our previous responses (both separate and general responses) for more detailed discussions.
>
> ---
>
> > **Q: How does HMN scale to more complex datasets?**
>
> **A:** We now included the evaluation results on ImageNet-10 in Table 2. Our additional evaluation also shows that HMN also scales to complex datasets like ImageNet-10 and achieves better performance.
>
> > **Q: The evaluation of HMN has a different setting. The training with HMN uses the cosine annealing LR scheduler, but the multi-step LR scheduler is more commonly used.**
>
> **A:** We report the numbers of training with HMN with a multi-step LR scheduler. We find the difference due to the LR scheduler choice to be marginal, and **the results with the multistep LR scheduler do not change the findings and conclusions of our evaluation.
>
> > **Q: HMN seems to have a larger number of GIPC than conventional methods using images as data containers, which can increase the training time.**
>
> **A:**  The reviewer's larger point is valid in that a larger GIPC size can impact training time. We would like to point out that this is a common feature of data parameterization methods, including IDC, HaBa, LinBa, and HMN. For example, LinBa generates 115 images per class with the 1 IPC setting for CIFAR10. However, data parameterization methods also achieve better accuracy. More GIPC can be treated as a tradeoff for better accuracy of data parameterization methods. We believe that this observation presents a promising direction for future research.
>  We leave the challenge of reducing GPIC for data parameterization methods, while retaining accuracy, for future exploration.
>
> > **Q: The baseline number of IDC is inconsistent with the numbers reported in the original paper.**
>
> **A:**  We thank the reviewer for pointing out this mismatch. There turned out to be some typos in our paper, and we now correct all the numbers. We would like to note that **the correction of those typos does not change the conclusion of our evaluation. HMN still achieves better results.**
>
> ----
>
> We have already revised the paper to include these points in the paper. **We also kindly refer you to our previous responses (both separate and general responses) for more detailed discussions.** We would like to check with you if our response and revised paper addressed your concerns and if we can provide more information to better address your concerns and clarify the claim in the paper!

---

> ### Author Response · Authors · 2023-11-23
> **A gentle reminder that our response is ready.**
>
> Dear Reviewer #3 (RRfc),
>
> We greatly appreciate your time and efforts in reviewing our paper again. Since the discussion period is about to end, we would like to gently double-check if our responses and revised paper have addressed your concerns. We are happy to provide more information if necessary!
>
> Thanks,
>
> Paper 4169 Authors

---

> > ### Comment · Reviewer_RRfc · 2023-11-23
> >
> > Thanks for your reply.
> >
> > However, I think the evaluation time of HMN is much longer than DC approaches such as DSA or even IDC.
> > Let's use CIFAR-100 with 50 IPC as an example. As shown in Table 5, the GIPC is 673 (#Instance-level memory), meaning that we generate 673 images for one class. However, CIFAR-100 only has 500 images per class. Hence, in the evaluation phase, the training time of HMN is longer than using the original images. Moreover, HMN achieves lower performance than using the original images. The only advantage of HMN I can see is that we can reduce the storage size. Why don't we compare using the same number of optimization steps ? (IDC reduces the number of epochs so that they can fairly compare to DC/DSA when considering the same training time)

---

### Official Review · Reviewer_Y8Kv · 2023-11-01

**Soundness:** 3 good
**Presentation:** 3 good
**Contribution:** 3 good
**Rating:** 6
**Confidence:** 2

**Summary:**

This paper works on the task of dataset condensation, i.e., producing a data generator whose generated data can be used to train a network that retain similar performance with respect to using a full dataset. The authors proposed a novel hierarchical data generation pipeline, which starts from a dataset level feature, then go through class-specific feature processing layers, and instance-level feature processing layers, and a uniform decoder to decode the features into images. Experiments show the proposed method achieves good performance on 3 benchmarks, on-par or out-performing methods with more expensive computes.

**Strengths:**

- The task of dataset condensation is an important and interesting task. This paper provide a good summarization of existing literature, and make solid progresses to this field.

- The proposed methods of hierarchical memory decoding is intuitive and makes a lot of sense to me. The followup pruning technique is also clean.

- Experiments show the proposed method work well on different benchmarks with different data budgets, outperforming existing methods which use more computes during training.

**Weaknesses:**

- The authors highlighted that they are using a more efficient training loss "batch-based", and claim it is sufficient to outperform a better but more expensive training loss "trajectory-based". Can the proposed method also use "trajectory-based" loss to further improve the performance? Or is it the proposed method itself is heavier then others, so that it can't be optimized using the better loss? Note experiments are optional in the rebuttal.

- As a researcher not working on this field, I had a hard time understanding what "IPC" means in experimental setup. Through searching other papers I get it is "Images allocated Per Class". It would be good if the paper can be more self-contained, to introduce this setting in the paper, and provide background knowledge on why this is used as the main setting. A more intuitive setting in my mind (without knowing the literature) is control the number of parameters of the learned data generators. Is this the same as the setup used in the paper?

**Questions:**

Overall this paper proposed a novel and valid method on an important task, with solid results. My questions are mostly for clarification, as I don't work on this field. I am happy to vote for an accept for now, conditioned on the authors clarifying my confusions in paper weaknesses during the rebuttal.

---

> ### Author Response · Authors · 2023-11-17
> **Response to Reviewer #2 (Y8Kv) (Part 1)**
>
> We thank Reviewer #2 (Y8Kv)’s time and effort for the reviews and highlighting some points in our paper that required further clarification. We address the raised questions and concerns below:
>
> **Q1:** The authors highlighted that they are using a more efficient training loss "batch-based", and claim it is sufficient to outperform a better but more expensive training loss "trajectory-based". Can the proposed method also use "trajectory-based" loss to further improve the performance? Or is it the proposed method itself is heavier then others, so that it can't be optimized using the better loss? Note experiments are optional in the rebuttal.
>
> **A1:** We thank the reviewer for bringing the scalability of different losses up for discussion, and we are happy to explain more about this.
>
> The major reason that we do not choose a trajectory-based loss is that trajectory-based loss is not scalable enough **not only with HMN (ours) but also with other data parameterization methods, like HaBa and LinBa.**
>
> In Table 7 (Appendix D.2), we discussed the memory usage of different parameterization methods. We find that HaBa with a trajectory-based loss consumes 48GB of memory even on the CIFAR10 dataset, and LinBa always exceeds the memory limitation. LinBa’s official implementation handles the memory issue by offloading GPU memory to CPU memory. However, CPU memory offloading leads to substantial computational overhead. For instance, in the CIFAR10 1 IPC task, LinBa requires 14 days to finish training on a 2080Ti, but training HMN with a batch-based loss takes only 15 hours with the same hardware. We thus choose the batch-based loss to train HMN.
>
> Besides, to have a fair comparison, we compare HMN with other data parameterization methods with the same batch-based loss in Table 3 ( Section 4.2 ), and we find HMN outperforms other methods.
>
> To sum up, training with a batch-based loss allows us to evaluate our methods on more complex datasets, like Tiny-ImageNet and ImageNet subset (see general response Q2), with reasonable resources and in a reasonable time. We agree with the reviewer that combining trajectory-based loss with HMN could be a line of future investigation if the hardware supported much higher GPU memory levels, much higher running times were acceptable (with CPU memory offloading), or more scalable trajectory-based methods were developed.  We have revised Appendix D.2 to include more discussion on training with a trajectory-based loss.

---

> > ### Author Response · Authors · 2023-11-17
> > **Response to Reviewer #2 (Y8Kv) (Part 2)**
> >
> > **Q2:** As a researcher not working on this field, I had a hard time understanding what "IPC" means in experimental setup. Through searching other papers I get it is "Images allocated Per Class". It would be good if the paper can be more self-contained, to introduce this setting in the paper, and provide background knowledge on why this is used as the main setting. A more intuitive setting in my mind (without knowing the literature) is control the number of parameters of the learned data generators. Is this the same as the setup used in the paper?
> >
> > **A2:** We are happy to provide additional clarification regarding the unclear representation in the paper.
> >
> > Reviewer #2 is right that IPC stands for the number of parameters used in the data containers. The abbreviation IPC stands for “Image Per Class”. This metric is used to measure how many tensor storage budget is equivalent to in terms of the number of images (same as SOTA works [1,2]). For example, 1 IPC for CIFAR10 stands for the storage budget:  pixels of an image * IPC * class = 3 * 32 * 32 * 1 * 10 = 30720 tensors. Similarly, 10 IPC for CIFAR100 stands for the storage budget:  pixels of an image * IPC * class = 3 * 32 * 32 * 10 * 100 = 30720000 tensors. Same to the previous works, we treat those tensors as float32 tensors. We added the definition of 'IPC' the first time it was mentioned (Section 1) in the revised version.
> >
> > There is a historical reason for using IPC as a metric. Before data parameterization is proposed, DC methods usually use images as data containers to store the training information, which causes IPC to naturally become a storage budget metric. In this paper, we follow the setting of the previous DC works and use IPC to measure storage budgets for a more convenient comparison.
> >
> > -------------
> >
> > We thank Reviewer #2 for the insightful reviews and for highlighting areas in our paper that required further clarification and discussion.  Those suggestions indeed strengthen the quality of our work. We hope that our response and revisions have successfully addressed your concerns and questions. We kindly hope that you can reconsider the rating in light of these updates. We are happy to have any further discussions or answer other questions that may arise, as we greatly value your feedback in this process.
> >
> > --------
> >
> > [1] Liu, Songhua, et al. "Dataset distillation via factorization." Advances in Neural Information Processing Systems 35 (2022): 1100-1113.
> >
> > [2] Deng, Zhiwei, and Olga Russakovsky. "Remember the past: Distilling datasets into addressable memories for neural networks." Advances in Neural Information Processing Systems 35 (2022): 34391-34404.

---

> ### Author Response · Authors · 2023-11-22
> **Thanks to Reviewer #2 (Y8Kv) and the Response Summary.**
>
> Dear Reviewer Y8Kv,
>
> We greatly appreciate your time and efforts in reviewing our paper, and **we would like to briefly summarize the response to your major concerns for your convenience.** We also kindly refer you to our previous responses (both separate and general responses) for more detailed discussions.
>
> ---
>
> > **Q: Can we combine HMN with a trajectory-based loss? How is the scalability of the proposed method?**
>
> **A:** In Table 7 (Appendix D.2), we show that trajectory-based loss is not scalable enough **not only with HMN (ours) but also with other data parameterization methods, like HaBa and LinBa (See original response for more details).** Training with a batch-based loss allows us to evaluate our methods on more complex datasets, like Tiny-ImageNet and ImageNet subset.
>
> In Table 7, we also show that our proposed end-to-end data condensation methods are more scalable than other SOTA data parameterization methods while achieving better performance. Our additional evaluation also shows that our proposed method also scales to complex datasets like ImageNet-10 and achieves better performance.
>
> > **Q: What is the definition of IPC?**
>
> **A:** The abbreviation IPC stands for “Image Per Class”. This metric is used to measure how many tensors the storage budget is equivalent to in terms of the number of images. For example, 1 IPC for CIFAR10 stands for the storage budget: pixels of an image * IPC * class = 3 * 32 * 32 * 1 * 10 = 30720 tensors.
>
> ----
>
> We have already revised the paper to include these points in the paper. **We also kindly refer you to our previous responses (both separate and general responses) for more detailed discussions.** We would like to check with you if our response and revised paper addressed your concerns and if we can provide more information to better address your concerns and clarify the claim in the paper!

---

> > ### Comment · Reviewer_Y8Kv · 2023-11-23
> > **Thank you.**
> >
> > Thank the authors for providing the rebuttal. Both my questions are now clarified. I keep my positive rating.

---

> > > ### Author Response · Authors · 2023-11-23
> > > **Thanks for your support.**
> > >
> > > We are happy to see that our response addresses your concerns. Thanks for your support on our paper.
> > >
> > > Paper 4169 Authors

---

### Official Review · Reviewer_mnZx · 2023-11-01

**Soundness:** 3 good
**Presentation:** 2 fair
**Contribution:** 3 good
**Rating:** 6
**Confidence:** 3

**Summary:**

This paper proposes a new method called Hierarchical Memory Network (HMN) that stores condensed data in a three-tier structure that reflects the hierarchical nature of image data. The authors exploited that HMN naturally ensures that generated images are independent and proposed a new algorithm to remove redundant images. The authors evaluated the model on four different datasets, and showed that their technique outperformed several SoTA baselines.

**Strengths:**

1- The paper proposed a new algorithm to store condensed data in a three-tier memory structure: dataset-level, class-level, and instance-level.
2- The authors proposed a pruning algorithm to prune redundant examples.
3- The authors demonstrated the effectiveness of their method on 4 different datasets. Their method outperformed several SoTA baselines by convincing margines.

**Weaknesses:**

1- It is difficult to follow some of the ideas presented in the paper. For example, the paper didn't mention the abbreviation for IPC. Also, the paper didn't talk about whether the decoder parameters is part of the budget or not.

2- The paper demonstrated the effectiveness of their method on tiny datasets, and they didn't address the scalability of their technique. For example, the generated images seem very pixelated and abstract, how does their method perform in a more complex settings.

**Questions:**

Table.1 What is the accuracy drop for randomly pruning 10% ?

Algorithm 1.  are you computing the accuracy in an online fashion, or you train the model for each subset selection? How this would scale for larger datasets?

---

> ### Author Response · Authors · 2023-11-17
> **Response to Reviewer #1 (mnZx)  (Part 1)**
>
> We thank Reviewer #1 (mnZx)’s time and effort for the reviews and suggestions to improve the paper. We address the raised questions and concerns below:
>
> **Q1:** 1- It is difficult to follow some of the ideas presented in the paper. For example, the paper didn't mention the abbreviation for IPC. Also, the paper didn't talk about whether the decoder parameters is part of the budget or not.
>
> **A1:** We are happy to clarify the unclear presentation in the paper. The abbreviation IPC stands for “Image Per Class”. This metric is used to measure how many tensors the storage budget is equivalent to in terms of the number of images(same as SOTA data parameterization work [1,2]).  For example, 1 IPC for CIFAR10 stands for the storage budget:  Pixels of an image * IPC * class = 3 * 32 * 32 * 1 * 10 = 30720 tensors. Similarly, 10 IPC for CIFAR100 stands for the storage budget:  Pixels of an image * IPC * class = 3 * 32 * 32 * 10 * 100 = 30720000 tensors. Same to the previous works, we treat those tensors as float32 tensors. We added the definition of 'IPC' the first time it was mentioned (Section 1) in the revised version.
>
> > Also, the paper didn't talk about whether the decoder parameters is part of the budget or not.
>
> Both the decoder networks and memory tensors count towards the storage budget calculation. We also mentioned this in Section 3.1 and at the end of Section 4.1. We further highlighted this point in the revised version to eliminate potential confusion for readers.
>
>
> **Q2:** 2- The paper demonstrated the effectiveness of their method on tiny datasets, and they didn't address the scalability of their technique. For example, the generated images seem very pixelated and abstract, how does their method perform in a more complex settings.
>
> **A2:** We thank Reviewer #1 for bringing up scalability for discussion.
>
> In Table 7 (Appendix D.2), we compare the scalability of end-to-end training algorithms of HMN with LinBa and HaBa, two data parameterization methods that have good performance. We find that HaBa with a trajectory-based loss consumes 48GB of memory even on the CIFAR10 dataset, and LinBa always exceeds the memory limitation. LinBa’s official implementation handles the memory issue by offloading GPU memory to CPU memory. However, CPU memory offloading leads to substantial computational overhead. For instance, in the CIFAR10 1 IPC task, LinBa requires 14 days to finish training on a 2080Ti, but training HMN with a batch-based loss takes only 15 hours with the same hardware. We believe that our end-to-end data condensation method not only achieves better performance than SOTA data parameterization methods but also is more scalable.
>
> Moreover, we also conducted the evaluation on ImageNet-10 (see General Response Q2). The results show that HMN still achieves strong performance on a more complex high-resolution dataset.
>
> > For example, the generated images seem very pixelated and abstract.
>
> We thank Reviewer #1 for pointing out the quality of the generated images. We think that it is a great question. We would like to clarify that the goal of data condensation is **not** to generate high-quality images, but **to generate images representing the training behavior of the original dataset**. Similar pixelation can also be found in other DC works, like DC, MTT, and HaBa. Pixelated images are fine as long as the resulting accuracy (performance) is still high for the level of compression desired.  We have revised Appendix D.6 to include generated image quality discussion.
>
> **Q3:** Table.1 What is the accuracy drop for randomly pruning 10% ?
>
> **A3:** We present the accuracy of random pruning on the CIFAR10 10 IPC HaBa condensed dataset as an addition to Table 1. We can see that, compared to AUM pruning, random pruning consistently has a larger accuracy drop, but 10% AUM pruning does not cause an accuracy drop, which indicates the existence of redundancy in training data generated by HaBa.
>
> | Pruning Rate |   0  |  10% |  20% |  30% |  40% |
> |:------:|:----:|:----:|:----:|:----:|:----:|
> |   AUM  | 69.5 | 69.5 | 68.9 | 67.6 | 65.6 |
> | Random | 69.5 | 68.7 | 67.5 | 65.9 | 63.8 |

---

> > ### Author Response · Authors · 2023-11-17
> > **Response to Reviewer #1 (mnZx) (Part 2)**
> >
> > **Q4:** Algorithm 1. are you computing the accuracy in an online fashion, or you train the model for each subset selection? How this would scale for larger datasets?
> >
> > **A4:**  In algorithm 1, we retrain the model for each subset selection for different hard pruning rate.
> >
> > However, Algorithm 1 is not a bottleneck with respect to running time. The time complexity of Algorithm 1 is proportional to the size of the condensed dataset (which Algorithm 1 prunes) rather than the original dataset. Since training on the condensed dataset is much faster than training on the original dataset, compared to the time required for data condensation, Algorithm 1 requires a relatively short computation time in practice. For example, data condensation with HMNs for CIFAR10 1 IPC needs about 15 hours on a 2080TI GPU, but the entire Algorithm 1 to prune the over-budget HMN only costs an additional 20 minutes.
> >
> > Moreover, we would note that Algorithm 1 runs only one time: we first condense an over-budget HMN; then, we use Algorithm 1 to identify the best hard pruning rate and prune the HMN to fit the budget. In other words, when we use HMNs for future training, we do not need to search for the best hard pruning rate again.
> >
> > Thank you for the question. We have revised Section 3.2.2 to further discuss the computational cost of Algorithm 1.
> >
> > ----------
> >
> > We thank Reviewer #1 again for the insightful reviews and for highlighting areas in our paper that required further clarification and discussion, which strengthens the quality of our work. We hope that our response and revisions have successfully addressed your concerns and questions. We kindly hope that you can reconsider the rating in light of these updates. We are happy to have any further discussions or answer other questions that may arise, as we greatly value your feedback in this process.
> >
> > --------
> >
> > [1] Liu, Songhua, et al. "Dataset distillation via factorization." Advances in Neural Information Processing Systems 35 (2022): 1100-1113.
> >
> > [2] Deng, Zhiwei, and Olga Russakovsky. "Remember the past: Distilling datasets into addressable memories for neural networks." Advances in Neural Information Processing Systems 35 (2022): 34391-34404.

---

> ### Author Response · Authors · 2023-11-22
> **Thanks to Reviewer #1 (mnZx) and the Response Summary on Major Concerns.**
>
> Dear Reviewer mnZx,
>
> We greatly appreciate your time and efforts in reviewing our paper, and **We would like to briefly summarize the response to your major concerns for your convenience.** We also kindly refer you to our previous responses (both separate and general responses) for more detailed discussions.
>
> ---
>
> > **Q: What is the definition of IPC? Is the decoder parameters part of the budget?**
>
> **A:** The abbreviation IPC stands for “Image Per Class”. This metric is used to measure how many tensors the storage budget is equivalent to in terms of the number of images. For example, 1 IPC for CIFAR10 stands for the storage budget: pixels of an image * IPC * class = 3 * 32 * 32 * 1 * 10 = 30720 tensors.
>
> Both the decoder networks and memory tensors count towards the storage budget calculation, which guarantees the fairness of the comparison to other DC methods.
>
> > **Q: How is the scalability of the proposed method?**
>
> **A:** In Table 7 (Appendix D.2), we show that our proposed end-to-end data condensation methods are more scalable than other SOTA data parameterization methods while achieving better performance. Our additional evaluation also shows that our proposed method also scales to complex datasets like ImageNet-10 and achieves better performance.
>
> > **Q: Does Algorithm 1. train the model for each subset selection? How this would scale for larger datasets?**
>
> **A:** Algorithm 1 retrains the model for each subset selection for different hard pruning rates. However, Algorithm 1 requires a relatively short computation time in practice. For example, DC with HMNs for CIFAR10 1 IPC needs about 15 hours on a 2080TI GPU, but the entire Algorithm 1 only costs an additional 20 minutes. Moreover, when we use HMNs for future training, we do not need to search for the best hard pruning rate again.
>
> ----
>
> We have already revised the paper to include these points in the paper. **We also kindly refer you to our previous responses (both separate and general responses) for more detailed discussions.** We would like to check with you if our response and revised paper addressed your concerns and if we can provide more information to better address your concerns and clarify the claim in the paper!

---

### Author Response · Authors · 2023-11-17
**General Response to All Reviewers (Part 1)**

Dear reviewers,

We thank all the reviewers for their constructive reviews and are very grateful for their feedback towards improving our work. We refer Reviewer mnZx as Reviewer #1, Reviewer Y8Kv as Reviewer #2,  Reviewer RRfc as Reviewer #3, and Reviewer UGPZ as Reviewer #4 for simplicity. (It is the order of the reviewers on the website.)

We are pleased that reviewers found our paper’s advantages: “The proposed method is novel” (All reviewers); “Show strong empirical performance” (Reviewer #1 and #2); “The data container pruning part is intuitive and novel” (Reviewer #1, #2, and #3);

For the comments and concerns discussed in the reviews, we write this general response to address some common concerns and separate responses to each individual review. Besides, we have revised the manuscript to improve the presentation of the paper and address the reviewer’s concerns as well.

-----

**Q1: (Clarification)** What is the definition of IPC?  (Reviewer #1 and #2)  Are decoder parameters a part of the storage budget? (Reviewer #1)

**A1:** The abbreviation 'IPC' stands for 'Image Per Class', a metric used to measure the equivalence of a tensor storage budget in terms of the number of images (same as SOTA works [3,4]). For example,  1 IPC for CIFAR10 stands for the storage budget:  Pixels of an image * IPC * class = 3 * 32 * 32 * 1 * 10 = 30720 tensors. 10IPC for CIFAR100 stands for the storage budget:  Pixels of an image * IPC * class = 3 * 32 * 32 * 10 * 100 = 30720000 tensors. Same in previous works, we treat those tensors as float32 tensors. We now include the definition of 'IPC' the first time it was mentioned (Section 1) in the revised version.

Both the decoder networks and memory tensors count toward the storage budget calculation for IPC. We have further highlighted this point in Section 4 in the revised version to eliminate potential confusion for readers.

**Q2: (Additional evaluation)** How is HMN scale in larger and more complex datasets, like ImageNet-10? (Reviewer #1 and #3)

**A2:**  We thank the reviewers for suggesting additional evaluation on a high-resolution dataset like ImageNet to better demonstrate the effectiveness of HMN. We now include the ImageNet-10 evaluation result in Table. 2 (Section 4.1) in the revised version for a more extensive comparison. We evaluated HMN on ImageNet-10 1 IPC setting and compared it with IDC (other data parameterization baselines are too resource-intensive to run).  The evaluation results are summarized below:

|      |  IDC |    HMN   |
|:----:|:----:|:--------:|
| 1 IPC | 60.4 | **64.6** |

From the table, we find that HMN still achieves strong performance on ImageNet-10, outperforming IDC.



**Q3: (Baseline number correctness)** Why are the results of IDC different from the reported numbers in IDC paper?  (Reviewer #3)

**A3:** We appreciate Reviewer #3 for pointing this out. There is a typo on the evaluation result for the IDC CIFAR10 50 IPC setting. We have corrected the number (The original number is the number of MTT.).
For the IDC CIFAR100 10 IPC setting, we use the data (44.8) from a data distillation survey paper [1]. However, we noticed that this number is different from the number reported in the IDC official repo [2]. We have updated this number to 45.1 in the revised version. After those modifications, the numbers in Tables 2 and 3 are corrected and the same as the author-reported numbers.

The numbers in Table 6 were the results that we got by rerunning the IDC evaluation.  Some differences can be caused by randomness in training. We now also update the numbers in Table 6 to author-reported numbers to guarantee consistency in the paper. We thank the reviewer for helping us correct this.

Furthermore, as suggested by Reviewer #4, we added a new baseline, IDM, to our comparison in the revised version.

We would like to note that **the correction of typos and adding a new baseline do not change the conclusion of our evaluation**. HMN still achieves better results.

---

> ### Author Response · Authors · 2023-11-17
> **General Response to All Reviewers (Part 2)**
>
> ## Revision Summary:
>
> We revised the paper based on the reviewers’ comments and suggestions. We also highlight significant changes in blue. Here is the summary of the major changes in the revised paper:
>
> Here is the summary of the changes that address reviewer comments in the revised paper:
>
> **Section 1 (Introduction)**: We added the definition of 'IPC' the first time it was mentioned.
>
> **Section 2 (Related work)**: We added the introduction of IDM.
>
> **Section 3.3 (Methodology)**: 1. We highlighted the design details of HMNs, including the shape of memories, the number of generated images per class, architectures of feature extractors and decoder, and referred them to Appendix for more details; 2. We highlight the discussion of the computational cost introduced by Algorithm 1.
>
> **Section 4 (Evaluation)**: 1. We updated typos of IDC numbers; 2. We added IDM as a new baseline for comparison.
>
> **Appendix B (Discussion)**: We added discussions on the cost of training with data parameterization methods and the inference cost of HMNs.
>
> **Appendix C.1 (Experimental Setting)**: 1. We added more details on the design of HMNs, like how many GIPC that HMNs have. 2. We include the evaluation setting for ImageNet-10.
>
> **Appendix D (Additional Evaluation)**: 1. We updated the IDC numbers in Table 6. 2. We added more discussion on the limitation of training with trajectory losses. 3. We added a new subsection Appendix D.4 to conduct an ablation study on different learning rate schedulers.
>
> **Appendix D (Visualization)**: We added a discussion on the quality of generated images by HMNs.
>
> -----
>
>
> Again, we thank the reviewers for their insightful feedback and suggestions, which largely help improve the quality and clarity of our paper. We hope that our clarifications, discussions, and additional evaluation address the reviewers’ concerns. We are happy to have any further discussions or answer other questions that may arise.
>
>
> ----
> [1] Sachdeva, Noveen, and Julian McAuley. "Data distillation: A survey." arXiv preprint arXiv:2301.04272 (2023).
>
> [2] https://github.com/snu-mllab/Efficient-Dataset-Condensation/
>
> [3] Liu, Songhua, et al. "Dataset distillation via factorization." Advances in Neural Information Processing Systems 35 (2022): 1100-1113.
>
> [4] Deng, Zhiwei, and Olga Russakovsky. "Remember the past: Distilling datasets into addressable memories for neural networks." Advances in Neural Information Processing Systems 35 (2022): 34391-34404.

---

### Author Response · Authors · 2023-11-21
**A reminder that our response and revised paper is ready.**

Dear all reviewers,

We greatly appreciate your time in reviewing our paper. We would like to kindly remind you that our responses and the revised paper are ready.

As the period of Author-Reviewer Discussions is approaching the end, we would like to check with you if our response and revised paper addressed your concerns. We are delighted to provide more information to better address your concerns and clarify the claim in the paper!

Paper 4169 Authors

---

### Meta-Review · Area_Chair_8cym · 2023-12-08

**Metareview:**

1) The paper proposes a new algorithm to store condensed data in a three-tier memory structure: dataset-level, class-level, and instance-level. They paper then shows how this dataset condensation works well on 4 datasets.
2) The reviewers agree  that the problem is important and the 3 tier memory structure is indeed novel, and that in the experiments provided their method does appear to work well compared to baselines.
3) All the reviewers bring up the lack of evaluation on larger more realistic datasets. While strong performance on 4 highly curated datasets like SVHN/CIFAR-10/CIFAR100 is a nice starting point. I believe at this point in ML it is important to move beyond these toy datasets to larger, less curated datasets where these dataset condensation techniques will be more important. Datasets such as YFCC-15M, CC-3M would be good starting points here.

I think the paper can be made much stronger with stronger evaluations on larger/less curated datasets which would make this a much more impactful paper (if this approach can condense 1B example datasets it would be *very* impactful). The authors do not need to run experiments on >1B example datasets to show this, they can take IID subsets of these large datasets and show their approach consistently condenses by a factor of 10 etc.) It's fine if the baselines they compare to don't evaluate at this regime, the given experiments show their approach outperforms these baselines anyway. I would strongly suggest that the authors look into less curated data mixes to condense. For these reasons I unfortunately do vote for rejection of this paper.

**Justification For Why Not Higher Score:**

Evaluations are only on very small scale datasets where condensation is not necessary. Paper can be improved with more thorough evaluation.

**Justification For Why Not Lower Score:**

N/A

---

### Decision · Program_Chairs · 2024-01-16

Reject